METHODS AND RESOURCES

# A toolkit for mapping cell identities in relation to neighbors reveals conserved patterning of neuromesodermal progenitor populations

Matthew French[1,2¤], Rosa P. Migueles[1,2], Alexandra Neaverson[3], Aishani Chakraborty[3], Tom Pettini[3], Benjamin Steventon[3], Erik Clark[3], J. Kim Dale[4], Guillaume Blin[1,2‡], Valerie Wilson[1,2‡], Sally Lowell[1,2‡*]

1 Institute for Stem Cell Research, School of Biological Sciences, The University of Edinburgh, Edinburgh, United Kingdom, 2 Centre for Regenerative Medicine, Institute for Regeneration and Repair, The University of Edinburgh, Edinburgh, United Kingdom, 3 Department of Genetics, University of Cambridge, Cambridge, United Kingdom, 4 School of Life Sciences, University of Dundee, Dundee, United Kingdom

‡ These authors are co-senior authors on this work.
¤ Current address: Department of Genetics, University of Cambridge, Cambridge, UK
* sally.lowell@ed.ac.uk

## Abstract

Patterning of cell fates is central to embryonic development, tissue homeostasis, and disease. Quantitative analysis of patterning reveals the logic by which cell-cell interactions orchestrate changes in cell fate. However, it is challenging to quantify patterning when graded changes in identity occur over complex 4D trajectories, or where different cell states are intermingled. Furthermore, comparing patterns across multiple individual embryos, tissues, or organoids is difficult because these often vary in shape and size. This problem is further exacerbated when comparing patterning between species. Here we present a toolkit of computational approaches to tackle these problems. These strategies are based on measuring properties of each cell in relation to the properties of its neighbors to quantify patterning, and on using embryonic landmarks in order to compare these patterns between embryos. We perform detailed neighbor-analysis of the caudal lateral epiblast of E8.5 mouse embryos, revealing local patterning in emergence of early mesoderm cells that is sensitive to inhibition of Notch activity. We extend this toolkit to compare mouse and chick embryos, revealing conserved 3D patterning of the caudal-lateral epiblast that scales across an order of magnitude difference in size between these two species. We also examine 3D patterning of gene expression boundaries across the length of *Drosophila* embryos. We present a flexible approach to examine the reproducibility of patterning between individuals, to measure phenotypic changes in patterning after experimental manipulation, and to compare of patterning across different scales and tissue architectures.

**Data availability statement:** Numerical data are provided as xl files on github alongside the code used in this study https://github.com/MattFrenchh/PRINGLE. https://doi.org/10.5281/zenodo.15802710.

**Funding:** This work was funded by the following grants: Wellcome Trust Senior Fellowship 220298 to SL from the Wellcome Trust https://wellcome.org/MRCgrantMR/S008799/1 to VW from the Medical Research Council https://www.ukri.org/councils/mrc/ BBSRC grant ref:BB/W002310/1 to GB from the Biotechnology and Biological Sciences Research Council https://www.ukri.org/councils/bbsrc/ WT ISSF3 award ref: IS3-R1.16 19/20 to GB from the Wellcome Trust https://wellcome.org/ EastBio PhD studentship BB/J01446X/1 to MF from the Biotechnology and Biological Sciences Research Council https://www.ukri.org/councils/bbsrc/ MRC grant MR/X018423/1 to JKD from the Medical Research Council https://www.ukri.org/councils/mrc/ Wellcome Trust Career Development Fellowship 227306/Z/23/Z to EC from the Wellcome Trust https://wellcome.org/ The funders had no role in the study design, data collection and analysis, decision to publish, or preparation of the manuscript.

**Competing interests:** The authors have declared that no competing interests exist.

**Abbreviations:** A-P, anterior-posterior; CLE, caudal lateral epiblast; CN, cell niche; CV, coefficient of variation; Dpp, Decapentaplegic; D-V, dorsal-ventral; EGF-R, Epidermal Growth Factor Receptor; FOV, field of view; HCR, hybridization chain reaction; NFI, normalized fluorescence intensity; NMPs, neuromesodermal progenitors; NR, Neighbor Ratio; NSB, node streak border; PBS, Phosphate-Buffered Saline; PS, primitive streak; PSM, presomitic mesoderm; SPs, somite pairs.

## Introduction

Quantifying patterning at single-cell resolution establishes when and where cell fate decisions are occurring, and helps clarify mechanisms by which cells make differentiation decisions. Impressive progress has been made in measuring and interpreting patterning of cell identity in tissues that exhibit relatively simple graded or striped patterns that can be readily visualized and quantified in 2D [1]. However, in many biological systems, cells change identity in relatively complex patterns across curved 3D shapes, making it challenging to quantify these patterning events. Comparing patterns between different species presents an additional challenge, particularly when species differ substantially in size and tissue morphology.

In recent years, improvements to segmentation have made it possible to measure marker expression at single-cell resolution within intact 3D tissues [2–7]. This opens up new opportunities for using neighbor relationships to quantify patterning in 3D [8,9]. For example, measuring differences between neighbors should make it possible to map complex graded changes across 3D space. Additionally, measuring local heterogeneity in cell identity provides clues about the logic of local interactions that coordinate cell fate changes. For example, particular types of spotty patterns can be generated by lateral inhibition [10] or stochastic cell fate allocation [11], while locally coherent cell identities are consistent with lateral induction [12], homeogenetic induction [13], quorum-sensing [14], or community effects [15–17]. Measuring changes in patterning after experimental manipulation of candidate regulators could then be used to identify mechanisms underlying local coordination of cell fate.

Approaches to measure patterning in a number of contexts have been developed [18–29]. However, challenges remain in measuring mesoscale patterning in cell identity during development [30]. This is in part because individual embryos differ from one another in size or shape, so there is a need for unbiased methods to define and normalize regions of interest that can be compared between different embryos [31,32]. Furthermore, defining different cell states, and relating the state of one cell to the states of its neighbors, can become particularly challenging when different cell identities merge along a continuum rather than falling naturally into discrete states. These problems are exemplified by the axial progenitor region in mouse embryos, which forms a complex curved surface with graded expression of known regulators of cell lineage, and has a well-defined fate map [33–35].

The axial progenitor region in E8.5 mouse embryos encompasses the caudal epiblast, which contains neuromesodermal progenitors (NMPs). These are bipotent progenitors that can generate neuroectoderm (producing spinal cord) and paraxial mesoderm (producing musculoskeleton) during axis elongation [36]. Cells exit from the NMP region in two ways. From the anterior edge of the NMP region they commit to neuroectoderm differentiation. Alternatively, they move towards the midline primitive streak, where they are committed to mesoderm differentiation, undergoing epithelial-to-mesenchymal transition and exiting towards the paraxial mesoderm. Differentiation of NMPs is governed by the activity of Wnt and FGF [36,37], but local cell-cell heterogeneity in cell identity [38,39] suggests that locally-acting feedback signaling between close neighbors may also contribute to controlling changes in NMP identity.

Here, we focus on the transition of NMPs towards a mesodermal fate, taking advantage of TBX6 as an early marker of mesoderm commitment [40–42]. By analyzing neighbor relationships, we identify steep local changes in TBX6 expression in the medial regions of the NMP progenitor region and show that TBX6-positive cells in this region tend to be surrounded by TBX6-negative cells and organized in a non-random distribution.

Notch signaling coordinates cell fate decisions between neighboring cells in many contexts, often acting by lateral inhibition [10] to generate local heterogeneity in cell identity but sometimes mediating lateral induction [12] to ensure local coherence of cell identity. Notch signaling favors mesoderm differentiation from hESC-derived NMPs at the expense of neural fate [43], positioning Notch as a likely candidate for generating heterogeneity in cell identity as NMPs differentiate into mesoderm during axis elongation. To test this, we extended our patterning analysis to embryos that had been subjected to Notch inhibition and found that local heterogeneity of TBX6 depends on Notch signaling.

To explore the broader applicability of our imagine pipeline, we examine local variability in cell identity during differentiation in monolayer and 3D cultures. We also measure scaling of NMP patterning between mouse and chick embryos, and examine scaling of gene expression boundaries across the anterior-posterior axis of *Drosophila* embryos.

We demonstrate that our image analysis pipeline enables spatial mapping of cell-cell variability and facilitates quantitative comparison of developmental patterning between embryos of different stages, different phenotypes, and even different species.

## Results

### Challenges for unbiased quantification of patterning

NMPs in the node-streak border and caudolateral epiblast are marked by coexpression of TBXT and SOX2 [36]. As NMPs commit to a mesoderm fate, they up-regulate TBX6 [40–42]. Therefore, the question of when and where NMPs become mesoderm-committed should in principle be answered by staining embryos for these three markers and analyzing the resulting images.

However, there are a number of challenges in interpreting such images. For example, TBXT and SOX2 exhibit graded distributions (Fig 1A and 1B), and their gradients do not appear to be aligned to a single axis (Fig 1A and 1B). This makes it challenging to objectively define an 'NMP containing region' based on visual inspection alone. Furthermore, the mesoderm commitment marker TBX6 does not seem to be upregulated at a clear boundary, but rather is expressed in a subset of cells within the caudolateral epiblast (Fig 1A and 1B: note contrast with the seemingly homogenous expression in the presomitic mesoderm). It is not easy to determine by eye whether these sporadic TBX6+ cells are restricted to a particular location or expressed in a particular pattern.

Additional challenges arise from the fact that embryos comprise multiple tissues arranged in 3D space: this can create confusion when embryos are viewed in 2D. In particular, the caudal epiblast (the focus of this study) lies on top of newly-formed mesoderm and has a complex curved 3D shape, making it difficult to visualize and quantify patterning in and around this region in a 2D representation (Fig 1A–1C). For example, TBX6 marks a layer of mesoderm that sits beneath the epiblast layer, but curves around to sit within the same optical plane (Fig 1B and 1C). Finally, there is the problem that different embryos have slightly different shapes and sizes, so it is challenging to compare quantitative measures of patterning between embryos. This becomes particularly important when assessing reproducibility of particular patterns, or when assessing patterning phenotypes after experimental manipulation of candidate regulators.

To address these problems, we sought quantitative computational methods to:

- Facilitate faithful 2D visualization of 3D patterning

- Visualize averaged patterning across multiple embryos of the same developmental stage

- Define regions of interest in a way that avoids arbitrary thresholding of individual markers

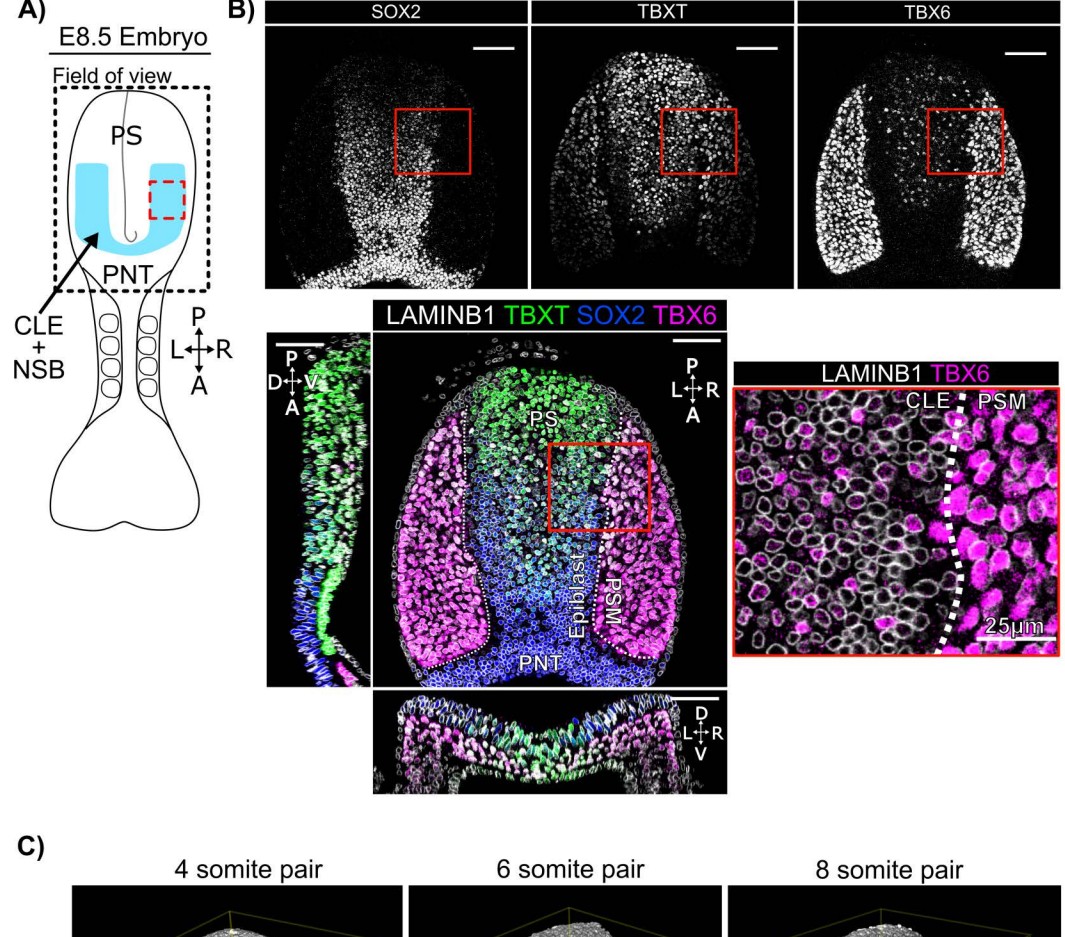

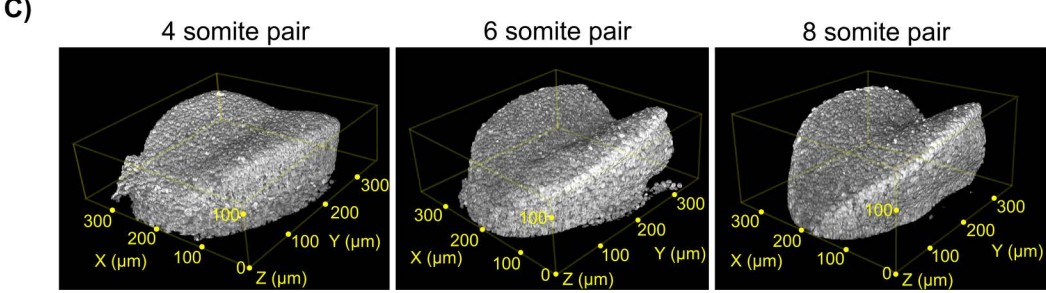

**Fig 1. Spatial patterning of TBXT, SOX2 and TBX6 in the posterior E8.5 mouse embryo. A:** Diagram representing a 4-somite pair E8.5 embryo and showing the positioning of the confocal imaging field of view (FOV) used in b and c as a dashed outline. The FOV encompasses the primitive streak (PS), and a domain comprising the neural and mesodermal bi-fated caudal lateral epiblast (CLE) plus node streak border (NSB) in blue. Anterior (A)/ Posterior (P) and Left (L)/Right (R) and posterior neural tube (PNT) are indicated. The dotted red box corresponds approximately to the insert shown in (b). The node is positioned at the posterior end of the primitive streak. **B:** Confocal z-slices across each plane of the field of view shown in (a) with insert (red box) showing the spatially heterogeneous TBX6 expression in the SOX2+TBXT+ CLE, and the spatially homogeneous TBX6 expression in PSM. White dotted line indicates the boundary between epiblast and the presomitic mesoderm (PSM). Unless specified, scale bars indicate 50 µm. **C:** 3D renders of LaminB1 signal at 4, 6, and 8 somite pair stage E8 embryos. Highlighting the increase in 'Pringle-like' epiblast curvature and the associated challenge posed when assessing patterning in 3D.

- Use neighbor-relationships to quantify and visualize the shape and steepness of gradients.

- Use neighbor-relationships to quantify and visualize fine-grained patterning.

- Compare patterning between different experimental conditions, or between different species.

PLOS Biology

## 3D Epiblast projection and alignment method: PRINGLE

Our first aim was to facilitate faithful visualization of 3D patterning in 2D, and to enable comparison of patterning between embryos. To achieve this, we developed a pipeline to computationally flatten the complex curved epiblast shape and to normalize the edges of the epiblast to defined landmarks.

We used a custom-written software package, PickCells, that combines tools for nuclear segmentation [3] with methods for quantifying the properties and positions of each cell within densely packed 3D tissues. In this pipeline, immunofluorescence confocal stacks are pre-processed to extract nuclear fluorescence intensity and neighborhood information (Fig 2A). In brief, nuclei are segmented using LAMINB1 nuclear envelope staining [3] to identify individual nuclei and to quantify the fluorescence intensity for SOX2, TBXT, and TBX6 immunofluorescence. The epiblast is manually isolated using PickCells software, and the nearest neighbors of each nucleus are identified in 3D using Delaunay triangulation (see Methods).

The epiblast is challenging to visualize in 2D because it curves in 3D convex and concave planes (similar to the shape of a Pringle potato crisp). We developed an algorithm which utilizes Projection and Relative Normalisation to aliGn muLtiple Epiblast (PRINGLE) (Figs 2B and S1). This has the effect of computationally flattening the curved epiblast, and makes it possible to unify multiple embryos by using relative positions of nuclei to landmark tissue structures, including the midline, node (manually labeled) (Figs 2B and S2), and epiblast edge (Figs 2B and S1). This provides absolute and relative positions of nuclei in relation to these landmarks. Using this approach, we visualized levels of SOX2, TBXT, and TBX6 immunofluorescence on individual PRINGLE projection epiblasts, as a starting point for further analysis (Fig 2C).

## Mapping the spatial distribution of Neuro-Mesodermal progenitors within the CLE

We next sought to map the NMP-containing region within and around the caudal-lateral epiblast as a starting point for asking when and where TBX6+ mesoderm arises in relation to this region.

Grafting of microdissected regions [34,35] has identified areas of the CLE that contain cells with both neural and mesodermal potency (black boxes in Fig 3A). However, this functional approach does not fully define the NMP region: for example, NMPs may also exist in some closely surrounding regions that were not included in grafts.

The NMP region can also be defined based on co-expression of SOX2 and TBXT [36,45]. However, this marker-based approach presents challenges because both TBXT and SOX2 are expressed in graded distributions, meaning that gating for positive cells can be somewhat arbitrary. For example, our manual gating of TBXT+SOX2+double-positive cells in E8.5 embryos (brown cells in Fig 3A) defines a region of double-positive cells that extends some distance beyond the known N-M-potent regions (Fig 3A). Furthermore, this analysis produces ill-defined borders between neighboring regions due to local heterogeneity in expression of TBXT and SOX2 (Fig 3A). We therefore sought an approach to harmonize the known N-M-potent regions with the SOX2/TBXT/TBX6 expression profile in the epiblast. Our aim was not to definitively assign NMP character to each individual cell, but rather to approximate a discrete region that is likely to contain cells of NMP identity.

We first quantified TBXT, SOX2, and TBX6 fluorescence immunofluorescence intensity for each nucleus in our segmented data (described in Fig 2). Then, values obtained from different images were normalized to enable cross-image comparison and smoothed using a moving average across local neighborhoods to reduce local noise (Fig 3B, Methods). Then, normalized and smoothed values of TBXT, SOX2, and TBX6 expression from multiple embryos were subjected to PCA analysis (Fig 3Ci). The pseudotime tool slingshot [44] was used to isolate a trajectory from SOX2+ to TBXT+SOX2+ to TBXT+TBX6+ to align nuclei along a TF pseudospace axis (Fig 3Cii), representing a continuum of cell states. Our aim here was not to define a temporal differentiation trajectory but rather to generate a range of values representing different cell pseudospace identities.

These values were then mapped back onto the "PRINGLE projection" epiblast (Fig 3D) in order to establish which pseudospace identities correspond to the regions of known NMP potency (black boxes in Fig 3D). On this basis, these values were assumed to represent an NMP-like identity (see Methods for more details). Nuclei that were defined in this

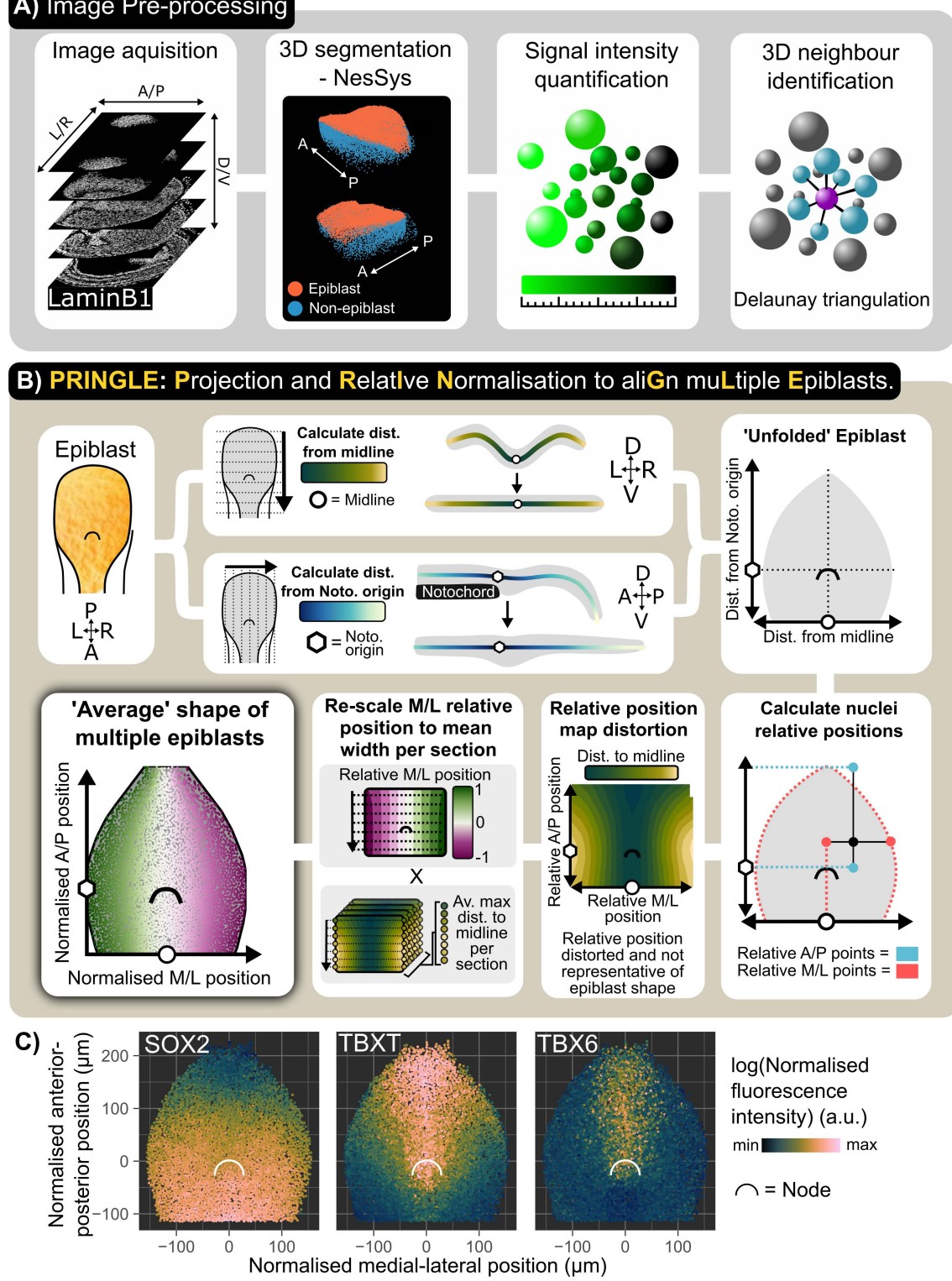

**Fig 2. 3D Epiblast projection and alignment method: PRINGLE. A:** Single cell analysis pipeline: individual nuclei are identified from 3D confocal images and manually labeled as epiblast or non-epiblast using the PickCells software. Fluorescence intensity is quantified within each nucleus and the neighbors of each cell in 3D are identified. **B:** Summary of PRINGLE method to project the 3D nuclei centroids in 2D and align multiple embryos: The

distance of nuclei centroids are calculated from the midline and notochord along principal curves in transverse and sagittal sections, respectively. These distances are then used as coordinates to project nuclei in 2D and create an unfolded epiblast. Unfolded maps generated from multiple embryos are then registered together by normalizing nuclei coordinates relative to tissue landmarks: the midline and edge of epiblast are used as landmarks for the left/right (L/R) axis, and the notochord origin and primitive streak posterior tip as landmarks for anterior/posterior (A/P) axis. However relative positions produce a distortion, therefore relative positions are normalized to the average epiblast width. This registration procedure results in an 'average' epiblast map in 2D. D: dorsal, V: ventral, A: Anterior, P: Posterior, M: Midline, L: Lateral edge of epiblast. This approach is explained in more detail in S1 Fig. **C:** A four-somite-pair epiblast showing SOX2, TBXT, and TBX6 expression (measured as outlined in (a)) mapped onto the average epiblast shape of multiple four-somite-pair epiblasts (established as outlined in (b)). Points represent nuclei centroids. Data for Fig 2 (C): Data file 1, https://doi.org/10.5281/zenodo.15802710.

way as having NMP-like identity mapped to the epiblast projection in a U shape that extends a short way beyond the known NMP regions, potentially encroaching on the neural-fated lateral border (Figs 3E and S3).

This approach provides a relatively unbiased method to define regions of interest based on combining information from multiple graded markers and relating this information to regions of experimentally determined fate. Comparison of the fate map with the pseudospace map allowed us to approximate a region in pseudospace that approximates to the NMP-containing region, providing a framework for defining patterning events in relation to this region.

## Mapping the shape and steepness of gradients of transcription factor expression in 3D

We next asked how patterning of SOX2, TBXT, and TBX6 relates to the position of the putative NMP region. This information is useful because, for example, steep changes in TF expression in time or space could indicate when and where cells are likely to be making differentiation decisions [46,47]. To measure graded changes in marker expression at high spatial resolution, we compared TF expression values between neighboring cells (Fig 4A: see S4 Fig and S1 Methods for a full description of this approach).

We first used the methods described in Fig 2 to computationally flatten and align epiblasts to make it possible to display averaged patterns across multiple embryos, visualized as manifold projections of the epiblast (PRINGLE projections). We then used the method described in Fig 3 to establish the putative NMP-containing region for each individual embryo (white U-shaped outlines in Figs 4B and S3). Embryos were staged according to the number of somite pairs (SPs), and the analysis described below is presented as the averaged pattern across multiple stage-matched embryos from 3 to 12 SP stages (Fig 4B).

Mapping the fluorescence intensity values onto averaged epiblast projections revealed the complex shapes of the gradients of SOX2, TBXT and TBX6 expression over time. SOX2 exhibits an M-shaped distribution, extending caudolaterally into the caudal lateral epiblast. Its expression shifts anteriorwards relative to the node between 4SP and 10SP stages. TBXT is highest at the posterior midline, decreasing anteriorly and laterally into the caudal lateral epiblast and anterior streak to form a V shape; this domain expands over time relative to the size of the epiblast, with the sharpness of the V shape flattening out somewhat at 10–12 SPs. TBX6 forms an elliptical expression domain centered on the middle of the primitive streak (Fig 4Bi)

We then asked where each marker exhibits the steepest change in expression. To quantify this, we calculated the magnitude of changes in smoothed fluorescence intensity values between a cell and its neighbors in 3D. (Figs 3Cii and S4). The resulting values were projected onto the manifold projection of epiblasts in order to visually highlight the regions with steepest changes in fluorescence intensity (Fig 4Bii). This analysis reveals that TBX6 expression changes sharply around the medial regions of the NMP region at all stages examined. TBXT expression also changes sharply in a similar region to TBX6 at SP 4,6 but expression flattens out somewhat by SP10–12. In contrast, SOX2 expression changes most prominently at the lateral and anterior edges of the NMP region (Fig 4Bii).

Finally, we used our pseudospace measure of state (Fig 3) to ask how fluorescence intensity values and local gradient steepness for each of the three transcription factors relate to cell identity (Fig 4C).

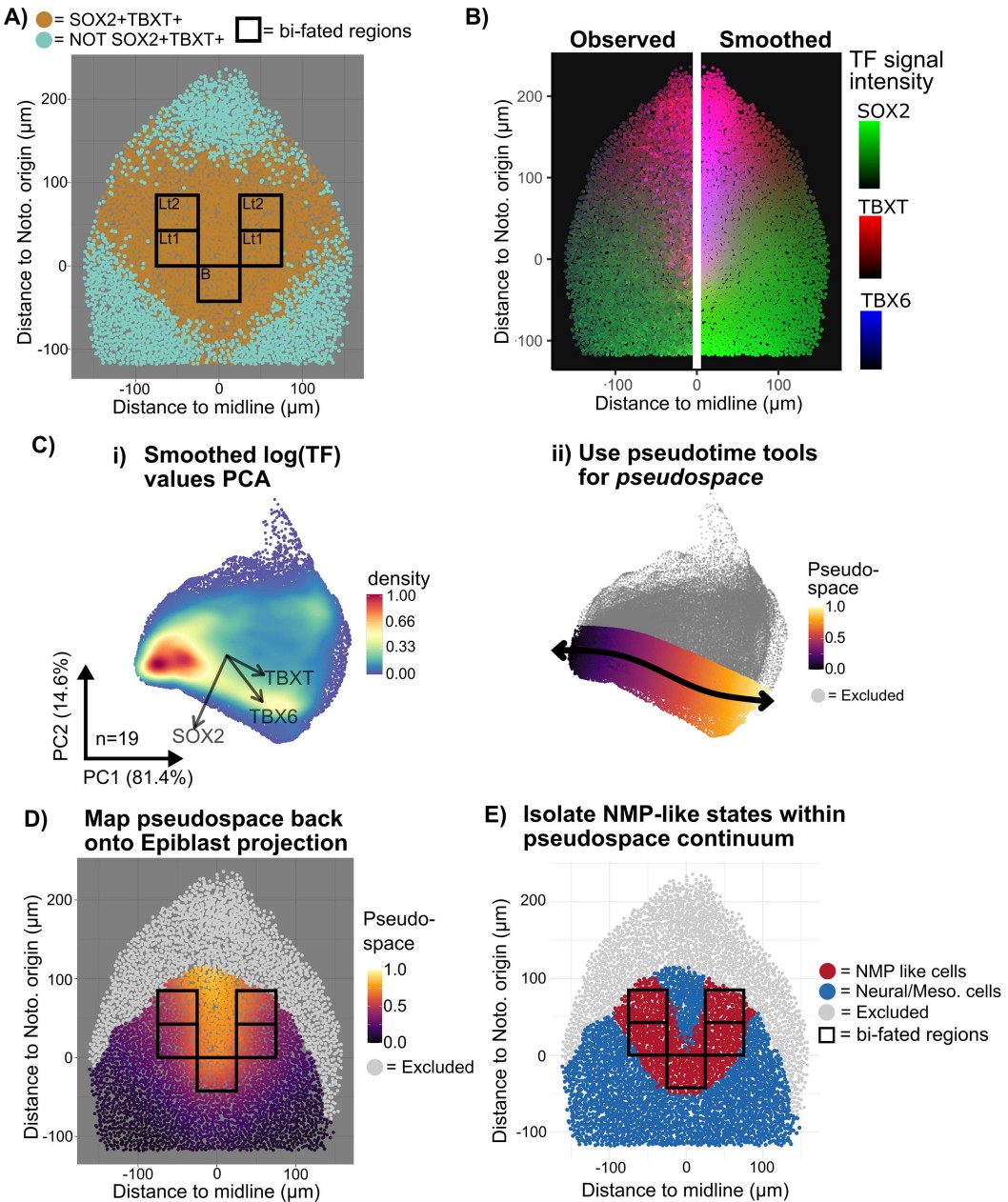

**Fig 3. Mapping the spatial distribution of Neuro-Mesodermal Progenitors. A:** Manifold projection of nuclei centroids in an example 4-somite-pair embryo showing SOX2+TBXT+ cells (orange) and the regions in the lateral epiblast (Lt1-2) and node-streak border (B) defined by grafting experiments [34,35] as having neural and mesodermal potency (bifated regions: black boxes). SOX2+TBXT+ are not restricted to the known N-M-bi-fated regions and instead map onto a broader spatial region. **B:** Iterative kernel averaging spatially smooths TF intensity values, shown by an example 4-somite-pair embryo. **C:** (i) PCA dimension reduction of the smoothed TF values in order to identify cell states. PCA loadings per TF indicated by arrow direction and magnitude (*n* = 19 embryos). (ii) The pseudotime tool Slingshot [44] identifies a path between the SOX2 high and the TBX6 high regions to create a pseudo-space continuum (*n* = 19 embryos). **D:** The pseudo-space continuum values (corresponding to cell states as defined by expression of TBXT, SOX2, and TBX6) mapped back onto the manifold projection of the four-somite-pair epiblast. Excluded cells (gray) are determined as described in the methods section. Pseudo-space values within known N-M-bifated regions are likely to correspond to NMP identities. **E:** The boundary between cells excluded because they fall outside the domain of potential neural/mesoderm fate (gray) and domains containing mono-fated, neural OR mesoderm cells (blue) is determined as described in the methods section. Red nuclei correspond to cells with pseudospace values defined in (Cii, D: see also main text) as corresponding to likely NMP identities ('NMP-like cells'). These map to a U-shaped region that extends beyond the boundaries of the known N-M-bifated regions (black boxes: based on functional grafting experiments [34,35]. Data for Fig 3 (A-E): Data file 1, https://doi.org/10.5281/zenodo.15802710.

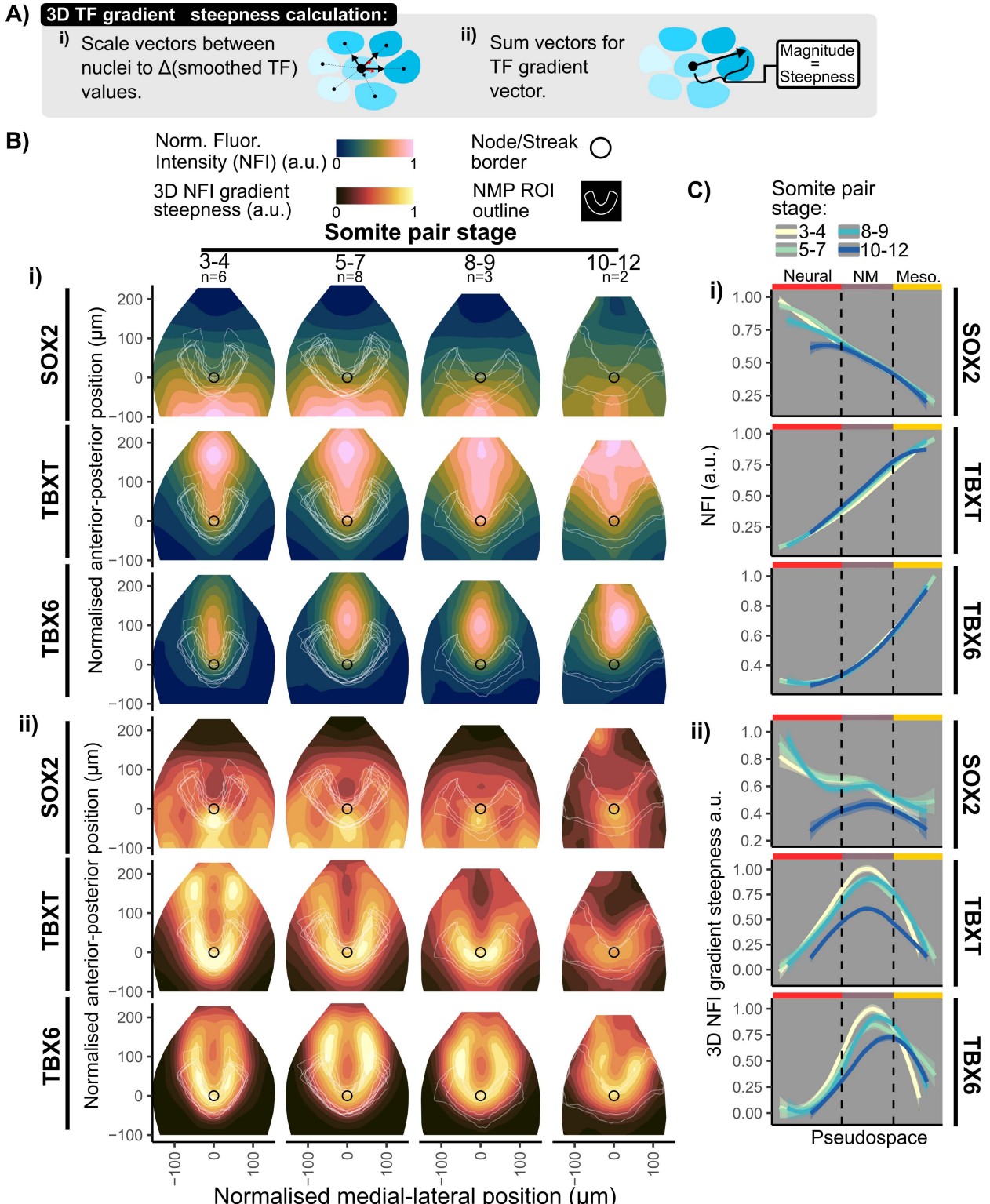

**Fig 4. Mapping the shape and steepness of gradients of transcription factor expression in 3D. A:** Method to map the shape of gradients across 3D space (i) First scale the unit vectors between a nuclei and its neighbors to the change in smoothed TF values. (ii) Then sum the vectors to obtain a single directional vector, with magnitude representing the steepness of the gradient. **B:** (i) Contour plots of average binned normalized fluorescence intensity

(NFI) measurements for SOX2, TBXT, and TBX6 in the normalized epiblast shape using PRINGLE, as described in Fig 2. Embryos are grouped per somite pair stage during E8.5, *n* numbers shown per group. White lines indicate the NMP region of interest (ROI) for each individual embryo as determined using the approach shown in Fig 3. (ii) Contour plots mapping TF gradient steepness map, calculated as shown in A. Averaged data from multiple embryos is shown, as explained in (c). Gradients of TBXT and TBX6 appear to be steepest in regions corresponding to NMP-like cells (NMP ROI) and gradients of SOX2 appear to be steepest in the node streak border. **C:** (i) Averages of single cell NFI measurements, as measured in Fig 2A, and measurement of gradient steepness, as measured in b (ii), per embryo and somite pair stage plotted in relation to pseudo space values, corresponding to cell identities based on integration of SOX2, TBX6, and TBXT expression as determined in Fig 3. This confirms that gradients of TBX6 and TBXT gradient are steepest within the NMP ROI and decreases into the PS, with TBX6 spatially lagging behind TBXT. Solid lines indicate the fit of a non-parametric multiple regression curve and shading indicates 0.05–0.95 confidence intervals calculated as (mean ± 1.96 * ($\sigma$/($\sqrt{n}$))). Vertical dashed lines indicate gating from fitting TF pseudospace to the N-M-bi-fated regions in Fig 3. Data for Fig 4 (B-C): Data file 1, https://doi.org/10.5281/zenodo.15802710.

In summary, we were able to use an approach based on local-neighbor-comparisons to map the steepest changes in SOX2 expression to a region where cells are committing to a neural fate, and to identify sharp changes in TBXT and TBX6 in regions where cells begin to adopt a mesoderm fate [34,35].

## Mapping the properties of cells in relation to their local neighborhood

Having explored graded changes of TF expression, we next examined mesoscale heterogeneity. In particular, we asked when and where cells tend to be different from their immediate neighbors (referred to here as "local variability"). This information is useful because, for example, the emergence of isolated TBX6+ cells within the epiblast could be consistent with sporadic mesoderm commitment in some cells followed by lateral inhibition of differentiation in neighbors [10].

We used the analysis described in Fig 2 (without smoothing between neighbors) to establish fluorescence intensity values for SOX2, TBXT, and TBX6 in each nucleus and to identify the nearest neighbors for each nucleus in 3D. We then used the coefficient of variation (CV) of TF values between a cell and its neighbors in order to obtain a simple relative measure of how similar each cell is to its local neighborhood (Fig 5A). This revealed that TBX6 exhibits the most local spatial heterogeneity, and SOX2 the least (Fig 5B).

We next asked whether high CV values map to particular embryonic regions. We used methods described in Fig 2 to computationally flatten and align epiblasts in order to display averaged patterns across multiple embryos, visualized as manifold projections of epiblasts. We then used the method described in Fig 3 to establish putative NMP regions for each individual embryo (white U-shaped outlines in Fig 5C). We then mapped CV values (Fig 5A and 5B) onto epiblast projections (Fig 5C). Embryos were staged according to the number of somites, and analysis is presented as the averaged pattern across multiple stage-matched embryos from 3 to 12 SP stages. (Fig 5C: note that these are the same embryos that are analyzed in Fig 4). This provides a map of where cells tend to be different from their neighbors (high CV) and where they tend to be similar to their neighbors (low CV).

This analysis confirmed that SOX2 exhibits low CV values across the region examined, while TBXT exhibits modest CV values within the NMP region and a hotspot of local heterogeneity around the anterior tip of the primitive streak. In contrast, TBX6 has high CV values mapping to the medial edges of the putative NMP region (Fig 5C). Plotting CV values for each cell in relation to pseudospace measure of cell identity (Fig 5D) confirms that local variability in TBX6 is associated primarily with the putative NMP region.

We then sought to distinguish between three patterning scenarios: a cell is surrounded by (1) higher, (2) the same, or (3) lower TF values compared to itself (Fig 5E). These scenarios can be distinguished by measuring the ratio between fluorescence intensity of each cell to the averaged fluorescence intensities of its neighbors (Neighbor Ratio: NR). In order to facilitate comparison between different cells, the log of NR is calculated to ensure proportional influence on NR for changes in numerator (cell) and denominator (average neighbor) (S5 Fig). To evaluate whether the observed NR values are consistent with a random distribution, synthetic data was produced assuming a normal distribution of TF values

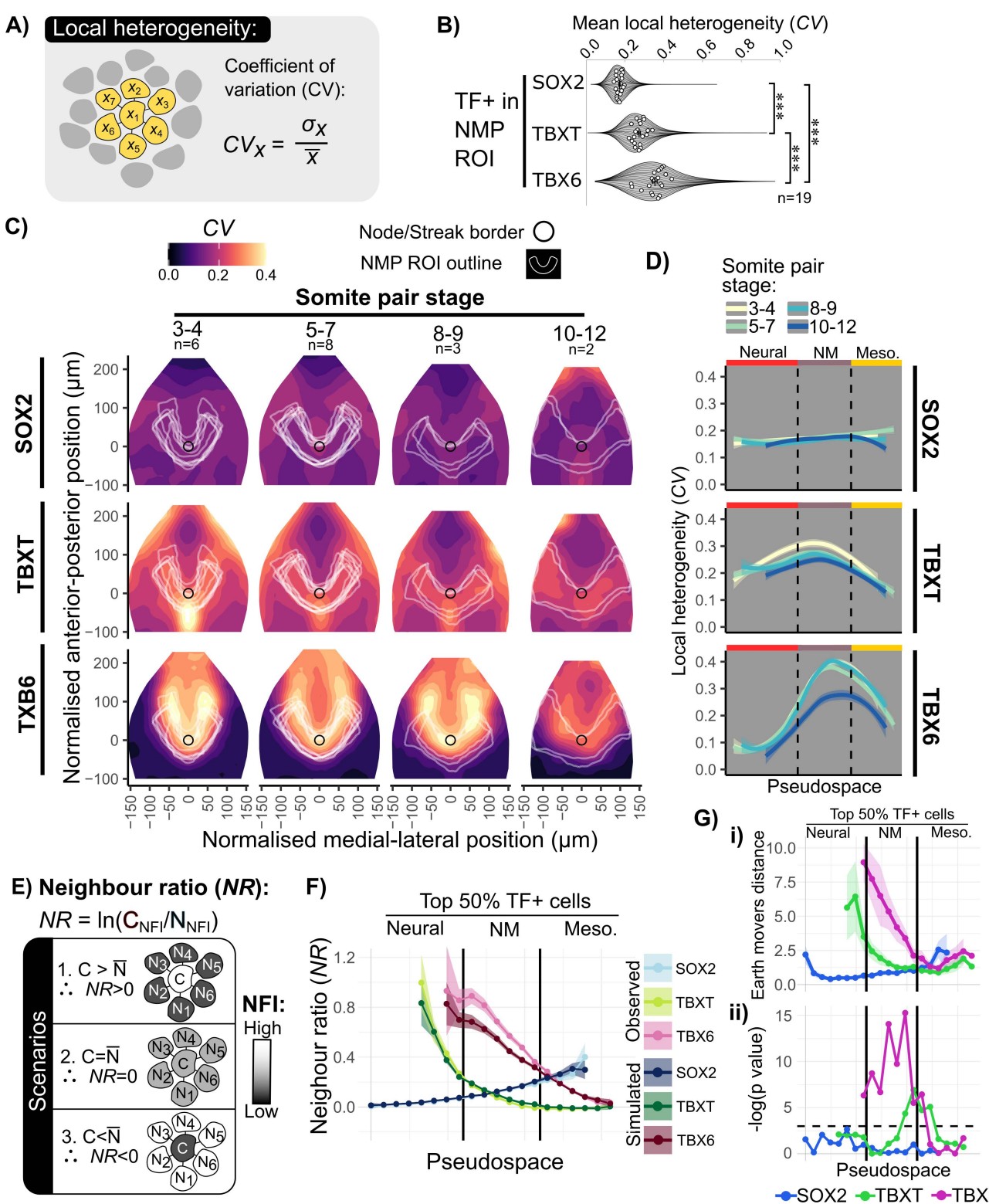

**Fig 5. Mapping the properties of cells in relation to their local neighborhood. A:** The coefficient of variation (CV) is used as a metric for the degree of local heterogeneity in fluorescence intensity values for a cell and its direct neighbors (in 3D), defined as the standard deviation divided by the mean of fluorescence intensity values for a cell and its neighbors. Neighbors are defined as in Fig 2A. **B:** Violin Superplots showing CV values for SOX2,

TBXT, and TBX6 within NMP-like cells as defined in Fig 3. Local heterogeneity (CV) in TBX6 is greater than TBXT, which is greater than local heterogeneity in SOX2. Dots indicate the mean for each of 19 embryos. Statistical tests performed with paired $t$ test and Benjamini–Hochberg post-hoc correction.. *** = $p < 0.001$. **C:** Maps of local variability in TF expression across the epiblast. Contour plots showing CV values mapped onto EpiMap. Data is averaged from multiple embryos as described in Fig 4). U-shaped white lines indicate the NMP ROI as measured in Fig 3. High local variability (CV) for TBXT and TBX6 is observed within the NMP ROI. **D:** Heightened TBXT and TBX6 CV to the NMP region and decreases into the mesodermal PS. Lines indicate the fit of a non-parametric multiple regression curve and shading indicates 0.05 and 0.95 confidence intervals calculated as (mean ± 1.96 * ($\sigma/(\sqrt{n})$)). **E:** The neighbor ratio (NR) metric can distinguish patterning scenarios where a cell is (1) surrounded by cells with an averaged expression that is higher expression than itself, (2) the same as itself, or (3) lower expression than itself. Defined as ln(Cell TF value/ Average Neighbor TF value). **F:** Comparing the 'high' cells in the TF+ populations (upper 50th percentile) for SOX2, TBXT, and TBX6 to synthetic data assuming a normal distribution for a cell's expression and its neighbors. This shows a decrease in NR, indicating TBX6+ high cells are surrounded by cells expressing lower TBX6 levels than expected within a random pattern. Points indicate average NR per embryo, shaded areas show confidence intervals of 0.05 and 0.95 calculated as (mean ± 1.96 * ($\sigma/(\sqrt{n})$)). Cells were binned into even intervals and only cells within these intervals were considered for confidence intervals and statistical summary metrics. **G:** Statistical analysis of the difference in observed and synthetic data NR values along pseudospace bins using (i) earth mover's distance between normalized distributions for each embryo and (ii) $p$-values resulting from one way ANOVA between observed and synthetic embryo means. The difference in TBX6 NR values is statistically significant in the NMP ROI with a high difference in distributions compared to TBXT or SOX2. Shaded areas show confidence intervals of 0.05 and 0.95. Dashed line in (ii) indicates $p$-value = 0.05. Data for Fig 5 (B–D and F–G): Data file 1, https://doi.org/10.5281/zenodo.15802710.

between a cell and its neighbors (S6 Fig). For example, observed and synthetic values for the top 50% TBX6+ cells were plotted for each cell in relation to their Pseudospace identity as determined in Fig 3 (Fig 5F: see S6 Fig for more details). We then computed the earth mover's distance [48] between the observed and synthetic NR distribution to obtain an informative measure of differences between these distributions (Fig 5G)

This analysis (Fig 5F) indicates that, within the putative NMP region, SOX2 and TBXT have relatively low NR values and that, at least in the case of Sox2, the local neighbor relationships seem consistent with a random distribution. In contrast, within the putative NMP region, this analysis indicates a non-random association of TBX6-positive cells with TBX6-negative neighbors.

This analysis method offers a quantitative approach to assess and compare local coherence versus heterogeneity in patterning of cell states. It confirms that SOX2+ and TBXT+ cells tend to be located in relatively coherent neighborhoods, i.e., they are mostly surrounded by cells of similar identity. In contrast, TBX6 exhibits considerable local heterogeneity within medial regions of the putative NMP domain, with isolated TBX6-high cells surrounded by TBX6-low cells. This raises the possibility that TBX6 expression is regulated by intercellular feedback signaling that enforces local cell fate diversification.

### Measuring local heterogeneity within embryos, gastruloids and monolayer differentiation

ES-cell-derived models of development are tractable experimental systems for studying cell-fate diversification but may not recapitulate all aspects of embryonic patterning. Having found that TBX6+ high tend to be surrounded by TBX6-low cells within the posterior epiblast of E8 mouse embryos (Fig 5), we asked to what extent this pattern is recapitulated in two commonly used in-vitro model systems: 2D monolayer differentiation of hES cells into NMPS [49,50] (Fig 6A), and 3D differentiation of mESCs into gastruloids [51,52] (Fig 6B).

Comparing patterning of embryos with patterning of cell-culture-based systems is challenging due to difficulties in identifying the particular populations in culture that correspond to a region of interest in vivo. For embryos, we used spatial cues to define the posterior epiblast and then defined the NMP region by combining information from marker expression with knowledge about potency based on grafting experiments (as described in Fig 2). For monolayer differentiation cultures, we used Chiron to impose an NMP character (S7 Fig) and consider the entire culture to be our region of interest. For gastruloids, we focused on the posterior region of the structure (red box in Fig 6B) and then identified NMP-like cells using an approach similar to that outlined in Fig 2, although in this case we relied exclusively on marker expression (S8 Fig) because the potency of particular subregions within gastruloids has not been defined.

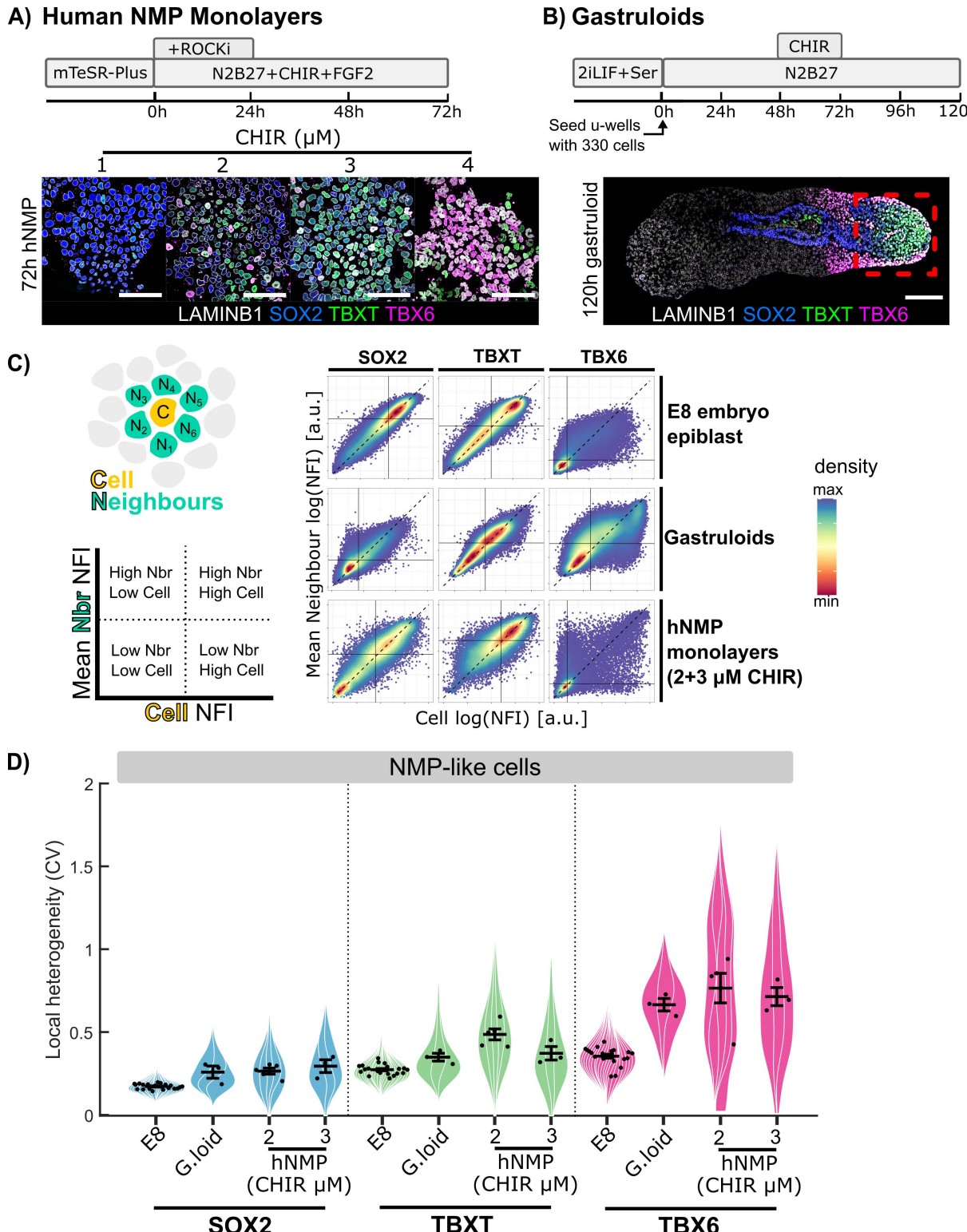

**Fig 6. Comparing local patterning between embryos, gastruloids and monolayer cell culture. A:** Protocol for generating hNMPs from hESC and representative confocal images of the culture fixed at 72 h and stained for SOX2, TBXT, TBX6, and LAMINB1 (further characterization S7 Fig). **B:** Protocol for generating gastruloids [51] with a representative confocal image of a gastruloid fixed at 120 h for SOX2, TBXT, TBX6, and LAMINB1. Red

box indicates typical field of view for high resolution image acquisition and subsequent neighbor analysis. **C:** Density plots comparing each nucleus's TF normalized fluorescence intensity (NFI) value to its neighbors shows strong linear correlations for TBXT and SOX2, but not for TBX6 with many 'high' TBX6 cells surrounded by 'low' TBX6 cells. Consistent in the E8 epiblast, gastruloids, and in vitro human NMP like cell monolayers. Black line indicates the median and the dashed line indicates $x = y$ line. **D:** Violin superplots of TF local heterogeneity (coefficient of variation [CV]) for respective TF+ cells within NMP-like cells isolated in gastruloids (process outlined in S8 Fig) and hNMP monolayers compared to in vivo WT E8 bi-fated region. NMPLC populations within in vitro models are more heterogeneous than comparative in vivo NMP populations. All scale bars indicate 50 μm. Points represent replicate means. Error bars represent 5%–95% confidence intervals. Gastruloids $n = 3$ with 3 technical replicates. Data for Fig 6 (C–D): Data file 1 and Data file 2, https://doi.org/10.5281/zenodo.15802710.

We confirmed that NMP differentiation from hESC was efficient at concentrations of 2 or 3uM CHIR99021 (Chiron) based on co-expression of TBXT and SOX2. (S7 Fig). We also confirmed successful emergence of NMP-like cells within the tip of gastruloids generated from mESC (S8 Fig). Interestingly, the NMP-like cells most resembling those defined in vivo (Fig 3B-3D) are located proximally to mesoderm cells at the tip of the elongated gastruloid (S8 Fig), reminiscent of the location of NMPs within the tail bud [35].

We then compared local variability of SOX2, TBXT, and TBX6 within both these in vitro systems to data obtained from the epiblasts of E8 embryos (embryo data was obtained as shown in Fig 5). This analysis differs from that shown in Fig 5 because our aim here was to measure overall levels of local variability in relation to cell identity, not to map CV values onto an averaged projection. We calculated normalized fluorescence intensity (NFI) values for SOX2, TBXT, and TBX6 in each nucleus and identified nearest neighbors in gastruloids and monolayer cultures using the same pipeline as for embryos (see Methods and Fig 2A). We then plotted the NFI for each cell against the mean NFI of all its neighbors in order to visualize how similar each cell is to its local neighborhood (Fig 6C).

This analysis (Fig 6C) confirms that, of the three TFs examined, TBX6 exhibits the most local spatial heterogeneity (many points do not align along the diagonal), and SOX2 the least spatial heterogeneity (most points align along the diagonal) in all three experimental systems. In the case of embryo and gastruloids, more cells lay beneath the diagonal than above it, indicating that TBX6-Hi cells tend to be surrounded by TBX6-low cells in both cases. This pattern was not recapitulated for monolayer differentiation cultures. To more carefully compare heterogeneity between different systems, we calculated the coefficient of variation (CV) of TF values between each cell and its neighbors (see Fig 5A). We then plotted the mean CV for each culture, gastruloid, or embryo (Fig 6D). This analysis confirms that hES-derived monolayer differentiation in unconstrained cultures exhibit a higher degree of local variability in TBX6 when compared with embryos or gastruloids. TBX6 heterogeneity in gastruloids was lower than in monolayer cultures but higher than in embryos.

We conclude that our analysis methods could be useful for comparing local variability between different experimental models, although a detailed like-for-like comparison would require knowledge of the fates of cell neighborhoods that is currently not available in cell culture model systems.

## Local heterogeneity in TBX6 is sensitive to Notch inhibition

One advantage of our analysis pipeline is that it allows quantitative assessment of changes in patterning after experimental manipulation of candidate regulators. It has recently been reported that Notch signaling can influence differentiation of hES-derived NMPs [43]. Notch mediates lateral inhibition to generate heterogeneity in many other contexts [10,43], and can regulate Tbx6 expression [53,54], making it an excellent candidate for contributing to the heterogeneity in TBX6 that we observe within the NMP region (Fig 5). We therefore asked if we could use our image analysis pipeline to detect changes in TBX6 patterning after pharmacological manipulation of Notch activity.

Posterior explants of E8.5 embryos were cultured with Notch signaling inhibitor LY or with vehicle-only (DMSO) for 12 h (Fig 7A). Fluorescence intensities were measured for SOX2, TBXT and TBX6. Embryos were analyzed using the approaches described in Figs 2–5 (Fig 7A and 7B). We confirmed that embryo explants displayed broadly normal tissue morphology, similar to a late E8.5 or early E9.0 embryo, after the short 12h culture period (Fig 7B), indicating that the

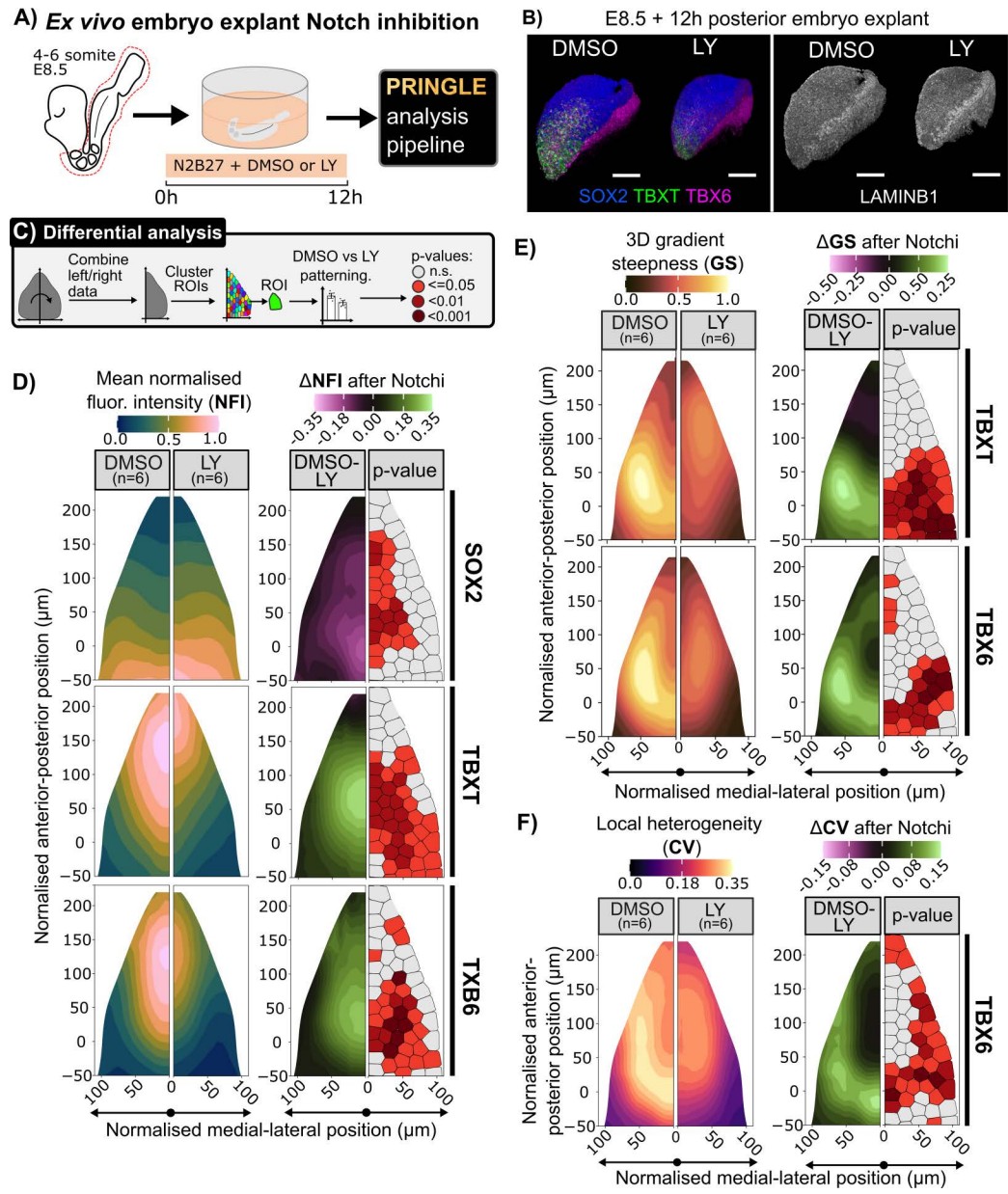

**Fig 7. Mapping changes in patterning after inhibition of Notch activity. A:** Culture method where a posterior explant of an E8.5 embryo is cultured in N2B27 media with the Notch inhibitor LY at 150 nM or an equal amount of DMSO for 12 h. **B:** Representative 3D renders of explants for each condition, stained for LaminB1 or SOX2, TBXT, and TBX6 indicating normal tissue morphology but a reduction in TBXT and TBX6 expression. Anterior (A), posterior (P), dorsal (D), ventral (V) axes are indicated. Scale bars indicate 100 μm. **C:** Analysis pipeline to evaluate the effect of Notch inhibition on patterning. After using PRINGLE to create an EpiMap, the epiblasts are folded at the midline and clustered with k-means to demark comparable ROIs. Patterning metrics are compared between DMSO and LY in these regions. One way ANOVA is conducted to identify the statistical significance of the difference of patterning metrics, regions are filled with associated colors to denote the p-value. **D:** Contour plots of TF normalized fluorescence intensity (NFI) projected onto the EpiMap with the difference in DMSO vs. LY conditions and associated p-values per ROI. Displaying an increase of SOX2 NFI in the PS and a posterior withdrawal of the TBXT and TBX6 domains in the Notch inhibited condition. **E:** Contour plots of NFI gradient steepness projected onto the EpiMap with the difference in DMSO vs. LY conditions and associated p-values per ROI. Note the overall decrease in TBXT and TBX6 gradient steepness, with a new posteriorly located nonlinear TBXT gradient. **F:** Contour plot of TBX6 local heterogeneity (CV) on EpiMap. Note the statistically significant decrease in the CLE and NSB like regions TBX6. Data for Fig 7 (D–F): Data file 3, https://doi.org/10.5281/zenodo.15802710.

overall shape and size of the epiblast region was not grossly affected by Notch inhibition (although we do not exclude the possibility of moderate effects on shape and size).

To visually and quantitatively compare control (DMSO) and Notch-inhibitor-treated (LY) epiblasts, we used the following procedure: First, we took advantage of the bilateral symmetry of the embryo and combined the left and right portions of our PRINGLE projections into one single projection (Fig 7C). This allowed us to provide a compact view of our data for side-by-side comparison between treated and control conditions (Fig 7D–7F).

Next, we used k-means clustering to partition projections into bins of approximately 100 cells (Fig 7D–7F). This binning approach allowed us to compute the statistical significance of differences observed between conditions and report the spatial distribution of p-values on the PRINGLE projection (Fig 7D–7F).

Control DMSO-treated embryos show a distribution of SOX2, TBXT, and TBX6 similar to that previously observed in 8–10SP E8.5 embryos (compare Fig 4Bi with Fig 7D). In contrast, after Notch inhibitor treatment, the SOX2 expression domain slightly expanded anteriorly while expression domains of TBXT and TBX6 were reduced (Fig 7D), with the most significant difference observed within lateral epiblast regions that would normally be expected to contain NMPs (see Fig 3). This is broadly consistent with a role for Notch in promoting mesoderm fate at the expense of neural differentiation [43].

We then measured the shape and steepness of TF gradients, as described in Fig 4. Notch inhibition brought about a significant decrease in the steepness of TBX6 and TBXT gradients within the more anterior regions of the CLE, consistent with a depletion in NMPs overall (Fig 7E). Finally, we measured local heterogeneity in TBX6 as described in Fig 5. Strikingly, although TBX6 was still expressed in the presence of Notch inhibitors, local heterogeneity was greatly reduced (Fig 7F).

We conclude that patterning of NMPs and their early mesoderm derivatives is sensitive to Notch inhibition. This showcases the utility of our patterning analysis as a useful means to determine the effect of developmental regulators on aspects of patterning, at tissue-scale and at the level of individual cells and their neighbors.

## Using PRINGLE to examine scaling of axial patterning between chick and mouse embryo

To demonstrate the utility of the PRINGLE pipeline across different vertebrate species, we adapted PRINGLE for use in chick embryos. The chick is a powerful experimental model for developmental and biomedical research [55] and both neural/mesodermal fate and SOX2/TBXT expression have been mapped at stages comparable to mouse [56–60]. The chick undergoes axial development using a similar underlying logic to the mouse, but with differing scale and morphology (Fig 8), making it difficult to visually assess similarities and differences in patterning between the two species.

Quantitative analysis of patterning in chick presented a related set of challenges to those encountered in mouse embryos (Fig 1). Although the chick epiblast is flatter than the mouse, it can undulate (particularly after processing) in a way that makes it difficult to extract clear quantitative information either from maximum projections or optical slices (Fig 8A). Therefore, computational flattening is needed in order to compare between embryos.

An additional challenge comes from the large number of cells in the chick epiblast, which is an order of magnitude larger than the mouse epiblast at a similar developmental stage: this makes it impractical to carry out manual classification of epiblast cells, as we had done in mouse embryos. Finally, we wanted to expand the general utility of our approach by using mRNA-based hybridization chain reaction (HCR; [61]) rather than protein-based immunofluorescence for detecting transcription factor expression.

We carried out HCR for *SOX2 and TBXT*. We also used HCR to examine *MSGN1,* a nascent and paraxial mesoderm marker [58] whose expression overlaps extensively with TBX6 [62]. We collected confocal stacks of four-stage HH6 chick embryos (Fig 8A), and segmented the resulting images using a peri-nuclear cytoplasmic mask (Fig 8B and 8C). We adopted a machine learning approach to classify the large number of epiblast cells (Fig 8D), which were then flattened using the PRINGLE pipeline to smooth the undulating folds of the chick CLE. This allowed us to compare averaged projections that highlight visual and quantitative aspects of both chick and mouse NMP regions.

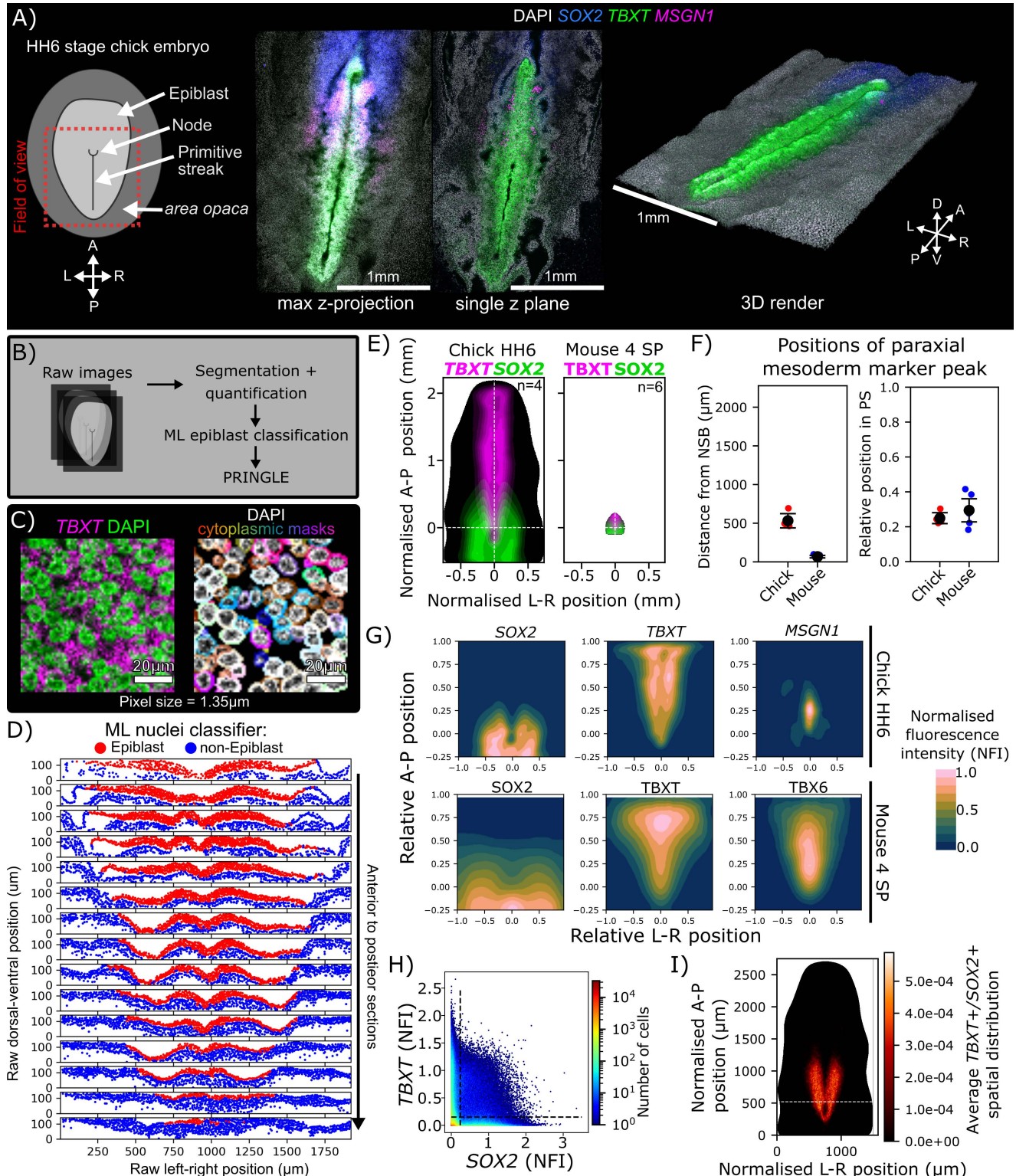

**Fig 8. Using PRINGLE to examine scaling of axial patterning between chick and mouse embryos. A**: Diagram of pre-somitic Hamburger-Hamilton stage 6 embryo, approximate field of view used in confocal imaging shown by red box (the developing head and notochord are not indicated). Confocal image stacks of HH6 embryo labeled with hybridization chain reaction (HCR) of *SOX2, TBXT*, and *MSGN1* transcripts represented with a

max projection, a single-z plane and 3D render, highlighting the difficulty in representing raw chick imaging data due to undulating nature of epithelium. **B:** outline of pipeline. **C:** Cytoplasm inference method for HCR quantification by peri-nuclear segmentation of DAPI signal. Nuclear segmentation performed with cellpose. **D:** Output of semi-supervised machine-learning classifier, with post-processing, to predict whether a nucleus is epiblast or non-epiblast, showing high accuracy. **E:** Size comparison of chick HH6 and 4-somite pair (SP) mouse epiblasts shown by PRINGLE projections to normalized A-P and L-R positions. Average *SOX2/TBXT* cytoplasmic signal for chick and SOX2/TBXT nuclear protein for mouse represented by colored contours (average of four embryos). White lines represent midline (vertical line) and position of node-streak border (intersection with horizontal line). **F:** The absolute and relative positions of peak paraxial mesoderm to the node streak border (NSB) and PS end show scaling of the focal point to ~30% of primitive streak length despite a large difference in absolute length. Peak paraxial mesoderm is defined as nuclei in the top 90th percentile for paraxial mesoderm marker expression, *MSGN1* in chick and TBX6 in mouse. Error bars show standard deviation. **G:** Comparison of relative global gradient maps for mouse and chick epiblasts show similar "M" shape for SOX2, TBXT "T" shape, and elliptical paraxial mesoderm pattern by TBX6 in mouse and *MSGN1* in chick. **H:** Density plot showing the pairwise expression of *SOX2* and *TBXT*, manual thresholds shown as black dashed lines outlining *SOX2+/TBXT+* population. **I:** Average density distribution ((no. cells per bin)/(total no. cells)) of *SOX2+/TBXT+* cells forms a u-shape around the PS, similar to mouse NMP region (Fig 3). White dashed line indicates the position of the node-streak border. Data for Fig 8 (D–I): Data file 4, https://doi.org/10.5281/zenodo.15802710.

Expression of *SOX2* and *TBXT* in chick was strikingly similar to that of mouse, despite a~10-fold difference in absolute size (Fig 8E). Furthermore, the distance between the node-streak border and the peak of expression of paraxial mesoderm markers TBX6 (mouse) or *MSGN1* (chick) is~10× greater in chick, although when scaled to the length of the primitive streak, the relative distances are indistinguishable (Fig 8F).

Comparing scaled chick and mouse PRINGLE projections showed graded expression of each transcription factor: chick *SOX2* expression formed an 'M' shape and *TBXT* a 'T' shape similar to mouse, while *TBXT* and *MSGN1* peaked at the midline (Fig 8G).

Finally, we mapped the putative NMP region in chick based on co-expression of *TBXT* and *SOX2* (Fig 8H and 8I). These data are in keeping with previous reports of the distribution of SOX2/TBXT double positive cells in chick embryos [58]. The SOX2/TBXT-coexpressing region forms a 'U' shape with the node at the lower end and the two sides extending into the anterior ⅔ of the CLE, similar to the proportion of the CLE that contains both neuromesodermal fate and SOX2/TBXT coexpressing cells (Fig 8I, compare with Fig 3).

We conclude that our pipeline for visualizing pattern within the epiblast can be extended to chick embryos, even though these differ considerably from mouse embryos in size and tissue architecture. This facilitates quantitative comparison between the two species, revealing conserved patterning scaled across the very different sizes of the mouse and chick epiblast.

## Using PRINGLE to analyze patterning in *Drosophila* embryos

To demonstrate the broader utility of the PRINGLE pipeline, we examined dorsal-ventral (D-V) positioning of gene expression boundaries across the anterior-posterior (A-P) axis of *Drosophila melanogaster* embryos. PRINGLE is well suited to this task because it is designed to quantify developmental patterning across multiple individuals relative to tissue landmarks, maintaining 3D information rather than relying on 2D optical slices.

We focus on the late blastoderm stage when mesoderm and neuroectoderm territories are becoming established. The nuclear concentration of maternally provided Dorsal (Dl) transcription factor at different thresholds ('high', 'medium', and 'low') provides spatial cues for the (D-V) axis. High Dl in the ventral region guides specification of presumptive ventral mesoderm, marked by *twist (twi)* and *snail (sna).* The adjacent neuroectoderm is marked by three longitudinal bands of homeobox gene expression: ventral nervous system defective *(vnd)*, intermediate neuroblasts defective *(ind)* at "medium" Dl, and muscle segment homeobox (*msh*) in 'low' Dl. A single cell row of *sim* marks the boundary between *twist* and *vnd.* [63,64] (Fig 9A).

The positioning of these domains has been extensively studied. Epidermal Growth Factor Receptor (EGF-R) signaling is confined to the 'medium' Dl region and promotes *vnd* and *ind* expression, while Decapentaplegic (Dpp) signaling is

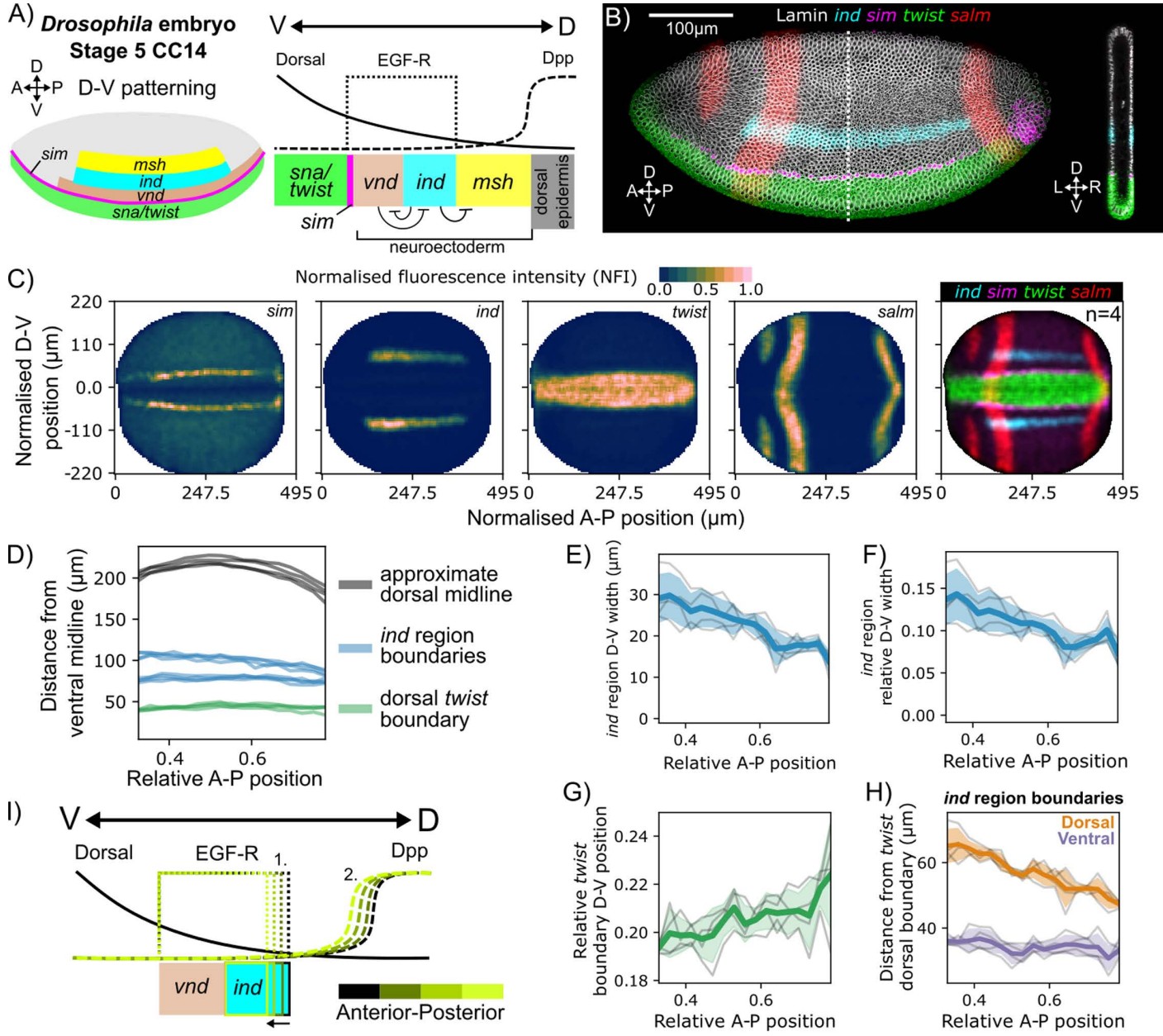

**Fig 9. Using PRINGLE to analyze patterning in *Drosophila* embryos. A**: Graphic outlining dorsal-ventral (D-V) patterning boundaries of neuroectoderm and presumptive ventral mesoderm in a late Stage 5 *Drosophila* embryo. Adapted from [53]. **B**: Confocal XY and XZ slices of representative embryo labeled by hybridization chain reaction (HCR) for *sim*, *ind*, *and twist*, with the anterior and posterior circumferential domains of *salm* marking the gnathal and tail regions, respectively [65]. Nuclear membrane marker Lamin labeled by immunofluorescence. White dashed line indicates the section position in the right panel. **C**: Output of PRINGLE projection from ventral midline after single cell quantification of HCR signal. *n=4*. **D**: The distance from the ventral midline to the *twist* boundary, *ind* boundaries, and approximate dorsal midline along anterior-posterior (A-P) axis reveals a seemingly stable *twist* boundary despite changes in the embryo D-V length. *ind* domain boundaries appear to converge at the posterior end. **E**: Directly measuring *ind* region width shows a linear reduction in width along the A-P axis. **F**: The linear reduction is maintained when considering width as a relative measure from the *twist* dorsal boundary to the dorsal midline. **G**: The location of the *twist* boundary relative to the ventral and dorsal midlines shifts dorsally along the A-P axis and could explain some but not all the reduction in *ind* width. **H**: In terms of distance from the *twist* boundary, the ventral boundary of *ind* is maintained at ~35μm but the dorsal boundary shifts ventrally along the A-P axis. **E–H**: Individual embryos shown by gray lines where left-right patterns were individually calculated and the average taken. Thick line represents the average across embryos. Shaded areas refer to 0.05 and 0.95 confidence intervals (mean ± 1.96 * ($\sigma$/($\sqrt{n}$)). **I**: Possible mechanisms to explain the maintenance of the ventral *ind* boundary with a variable dorsal boundary. Dorsal gradient and induced *vnd* does not change but **1.** the EFG-R active domain restricts and/or **2.** the Dpp activity expands to dorsally restrict *ind*. Data for Fig 9 (C–H): Data file 5, https://doi.org/10.5281/zenodo.15802710.

found in the 'low' DL region [64] (Fig 9A). The dorsal and ventral limits of each neuroectoderm band have been proposed to be differentially controlled with 'ventral dominance' of cross-repression [66]. For instance, the Dl and *twist* target *vnd* restricts ventral expression of *ind*, with dorsal repression of *ind* from Dpp downstream targets [67,68]. However, The Dl gradient displays the capacity to scale to different tissue sizes, but differential tissue and Dl scaling does not result in subsequent *ind* or *vnd* region scaling [69], suggesting that the regulatory mechanism controlling the *ind* boundaries is not fully understood.

We used PRINGLE to project 3D *Drosophila* ellipsoidal embryos into 2D, and quantify spatial boundaries of gene expression relative to tissue structures. We carried out HCR for *twist, ind, sim, and salm* (*salm* is included here to illustrate reproducibility of A-P patterning between embryos). Individual cells were identified using immunofluorescence for the nuclear marker Lamin. Multiview confocal image stacks were collected from four stage 5 cell-cycle 14 *Drosophila* embryos (Fig 9B) in order to examine reproducibility of patterning between four individuals.

Multi-view images were segmented with cellpose to identify individual cells [70] and computationally registered to select the side of embryo closest to the microscope objective (S9 Fig). PRINGLE was used to project the 3D blastoderm into 2D, using midline-marker *twist* to orient the embryos (Fig 9C). We then measured expression domains in the four embryos in relation to the boundaries of each embryo. This revealed that the dorsal limit of *twist* is constant over the A-P axis, despite a shorter absolute distance to the dorsalmost extent of the embryo at the anterior and posterior ends. In contrast, the region expressing *ind* appears to narrow posteriorly, but not anteriorly (Fig 9D).

The absolute width of the *ind* expression domain decreases towards the posterior of the A-P axis (Fig 9E). This reduction is also observed when considering width as a relative measure from the *twist* dorsal boundary to the dorsal midline (Fig 9F). The boundary of *twist* expression shifts dorsally (Fig 9G) as the dorsal boundary of *ind* shifts ventrally (Fig 9H). This supports the idea that the mechanisms that position the DV boundary of *ind* are modulated according to AP position, perhaps through interactions with EGF-R and/or Dpp (Fig 9I).

We conclude that our pipeline for measuring 3D patterning can be extended to fly embryos, even though these differ considerably from mouse or chick embryos in scale and tissue architecture, and can be used to explore how gene expression patterns scale to changes in embryo width across the axis of the fly embryo.

### Using PRINGLE to analyze patterning in *Drosophila* embryos

In overall conclusion, our image analysis toolkit can be used to examine reproducibility of patterning between individuals, phenotypic changes in patterning after experimental manipulations, and conserved and divergent features of patterning across different scales and tissue architectures.

### Discussion

Quantification of local patterning within tissues is a non-trivial task, even when analysis is restricted to 2D images [71]. Extending analysis to three dimensions presents additional challenges. Here, by measuring how similar each cell is to its local neighborhood in 3D, we could determine the local direction and steepness of graded transcription factor expression and extract quantitative descriptions of fine-grained heterogeneity in cell identity. We used these methods to examine when, where, and how changes in cell identity occur in the posterior of mouse embryos at the onset of axis elongation. We demonstrate the broader applicability of our pipeline by comparing patterning between mouse and chick embryos, and by analyzing the positioning of gene expression boundaries in *Drosophila* embryos.

It can be challenging to find robust ways to compare image data. Standardized micropatterns offer one solution to quantify reproducibility of patterning and to reliably measure the effects of experimental manipulations in experiments using cultured cell monolayers [72]. It is considerably more challenging to quantitatively compare patterning between embryos or 3D cultures because these have complex and somewhat variable shapes and sizes: for example, the E8.5 epiblast of the mouse embryo has a complex curved shape. This problem can be addressed with methods to project

complex curved shapes onto 2D planes [73]. Here we developed a simpler approach that is based on analysis of segmentation data rather than raw images, and which incorporates a module to normalize the spatial data to embryonic axes. This establishes a framework within which to quantify patterning across multiple embryos.

In order to compare local patterning between embryos, it is first necessary to define a region of interest within which patterns are to be compared. However, regions of interest are often not clearly demarcated by coherent expression domains with distinct boundaries. One such example is the region of the mouse embryo that harbors NMPs during axis elongation. In this study, we develop a method to integrate information from multiple graded markers with previous experimental data defining the fate of microdissected regions [34,35]. This establishes TBXT/SOX2 levels that define putative NMPs. Our approach here builds on a previous study [35] which used a simpler thresholding approach to define a U shape describing mid-level TBXT and SOX2 co expression at E8.5. Here we advance on that approach by avoiding arbitrary thresholding of individual markers.

The putative NMP domain we identify includes part of a region lateral to the node-streak border previously assumed to contain neural-committed progenitors. This may explain why this region shows two alternative contribution patterns: one, where cells exit en masse to the neural tube, and the other, where cells are retained in the tail bud and contribute to neuroectoderm and mesoderm: the dissected region contains cells of two alternate phenotypes: High SOX2/negligible TBXT neural-committed cells towards the anterior/lateral edge of the rectangular shape dissected for grafting, and mid SOX2/TBXT NMPs at the posterior/medial side. The grafting procedure partially disperses cells in these regions, which may have resulted in some grafts retaining NMPs while others contain only neural-committed cells. Significantly, we also find that coexpression of high TBXT/low SOX2 at the midline occurs in a mesoderm-committed region, suggesting that mesoderm commitment is associated with a threshold level of TBXT rather than extinction of SOX2.

Interestingly, the gradients of all three TFs are steepest at the anteriormost part of the NMP region near the node. This locale contains the longest-lived NMPs [34], and contacts the most posterior notochord progenitors ventrally. Heterotopic transplants of NMPs show that contact with the posterior notochord progenitors increases the retention of NMPs as progenitors in the tailbud, without compromising their ability to differentiate [35]. Furthermore, the posteriormost notochord progenitors are vital for axial elongation [74]. The presence of steep gradients of all three TFs in this location is consistent with an NMP population anchored via a small and stable niche, the posterior notochord, from which cells can readily exit towards both neurectoderm and mesoderm.

The mechanisms that regulate mesoderm commitment are not fully understood. Mesoderm differentiation is driven by Wnt and FGF signaling, yet these pathways appear to be active throughout the caudal epiblast [75–78], with no evidence for upregulation of activity at the primitive streak. Furthermore, Wnt and FGF are required for maintenance of progenitors as well as their differentiation into mesoderm [36,37]. It therefore seems likely that additional factors must influence mesoderm commitment. Notch is one promising candidate: Notch ligands and targets are upregulated at the midline, and Notch can regulate mesoderm commitments [37,43,79].

In keeping with the idea that Notch regulates mesoderm commitment, our neighbor-comparisons reveal that mesoderm-commitment marker TBX6 is expressed in a spotty pattern within the NMP region. Isolated TBX6+ cells are non-randomly surrounded by TBX6- neighbors, and this pattern is sensitive to pharmacological suppression of Notch activity. Based on these findings an attractive hypothesis is that lateral inhibition through Notch [10] balances commitment to mesoderm fate with preservation of a progenitor pool as cells approach the midline. However, we cannot exclude other mechanisms: local variability between neighbors does not necessarily indicate lateral inhibition of cell identity: it could represent bistability in stochastic cell fate allocation at threshold levels of signaling inputs [80], or it could be the consequence of migration of cells from an initially coherent domain [81]. It would be interesting to use emerging neighbor-labeling technologies [82–84] to test more directly whether, for example, TBX6+ cells suppress mesoderm commitment in their neighbors.

Our PRINGLE pipeline enables cross-species comparison even between embryos of very different shapes and sizes. We compare patterning of NMPs between chick [58] and mouse [35], revealing a striking similarity in NMP patterning between these two species. This opens up interesting questions about how patterning is accurately scaled across a difference in size of over an order of magnitude.

We further extend the scope of our work by showing that PRINGLE can be applied to the prolate-spheroid-shaped early *Drosophila* embryo. This makes it possible to examine how positioning of gene expression boundaries are scaled across the length of the embryo, taking into account information in all three dimensions rather than relying on optical slices. We reveal differences between the anterior and the posterior in dorso-ventral scaling.

Some limitations of our approach remain. Our analysis here is limited to only a few markers. There may be opportunities to analyze single-cell-resolution-spatial-transcriptomics using our pipeline, but we have so far not explored the challenges of implementing high-dimensional imaging data.

Our analysis also depends on reliable single-cell segmentation, which can be challenging in some experimental setups, although methods for segmentation are improving all the time [85,86]. We also note that our analysis of patterning does not yet reach its full potential. More complex patterns could be identified based on the neighbor-relationships that we measure using our approach, taking advantage of mathematical modeling to extract non-obvious levels of organization [87].

In summary, our patterning-analysis methods give new insights into fine-grained patterning of cell identity during early post-gastrulation mouse development, and into scaling of patterning across tissues and between different species. These approaches should be applicable to any system that is amenable to single-cell segmentation, and it will be particularly interesting to extend these approaches to single-cell-resolution spatial transcriptomics datasets. We present these approaches as a toolkit for studying patterning during development, homeostasis, regeneration or disease.

## Methods

### Mouse embryo dissection and culture

Mice were bred and housed in the Animal Unit of the Centre for Regenerative Medicine in accordance with the provisions of the Animals Act 1986 (Scientific procedures). Pregnant female mice were culled by cervical dislocation by the Animal Unit staff. The uterus was dissected from the mice and placed into M2 media at RT, whereafter the decidua and Reichert's membrane were removed using forceps.

For embryo culture, the anterior portion up to the first somite was removed while in M2 media. These were immediately transferred to N2B27 media supplemented with Pen/Strep antibiotics and either 150 nM LY411575 or an equal amount of DMSO. These were cultured statically in an incubator at 37 °C and 5% $CO_2$ for 12 h.

Mouse work was approved by the Edinburgh University Animal Welfare and Ethical Review Board, and performed under Home Office project licence number **PP8085667**.

### Chick embryo dissection and fixation

Shaver Brown chicken eggs (from Henry Stewart & Co, Norfolk, England) were incubated at 37 °C in a humidified incubator for ~26 h to obtain stage HH6 embryos. Eggs were opened with forceps and the embryo removed and placed in Phosphate-Buffered Saline (PBS, Sigma Aldrich) before dissecting from the vitelline membrane. Embryos were then fixed using 4% paraformaldehyde (Sigma Aldrich, 158127) in PBS, overnight at 4 °C.

### *Drosophila* husbandry and embryo fixing

'Wild-type' Oregon-R flies were maintained at 25 °C. Embryos were collected on apple juice agar plates, dechorionated using 50% sodium hypochlorite solution, and fixed for 30 min shaking with 4% formaldehyde in 0.5× PBS and 50% heptane. Embryos were devitellinized by shaking vigorously for 1 min in 100% methanol, then washed and stored in 100% methanol at −20 °C.

## Mouse ES and gastruloid culture

E14Ju09 ES cells [88] were routinely maintained in FCS (Gibco) with 100u/ml LIF (produced in-house) on gelatinized plates as described in [82]. Prior to gastruloid formation, ES cell culture media was additionally supplemented with 3 μM CHIR99,021 and 5 μM PD0325901 5 μl 1 μM (2i LIF FCS media) for at least 10 days in order to reduce spontaneous differentiation.

Gastruloids were generated as previously described [51]. 2i/LIF/FCS ES cells are washed in PBS before adding 0.05% Trypsin EDTA solution. After ES colony detachment, 5–10 volumes of fresh 2i/LIF/FCS were added to quench the Trypsin EDTA solution. The cells and media were transferred to a universal tube and pelleted by centrifugation at 300$g$ for 3 min. The media was aspirated, and the pellet was resuspended in cold PBS, making sure to create a single cell suspension. This PBS wash was repeated once more, then the cell pellet was resuspended in prewarmed N2B27 to create a single cell solution. The cells were then counted and then a solution of N2B27 medium with 8,250 cells/mL was made. 40 μL of this medium, containing ~330 cells, was plated into untreated u-bottom 96 wells and incubated for 48 h at 37 °C and 5% $CO_2$. At 48 h, 150 μL of N2B27 and 3 μM CHIR99021 (Axon) were added into the well, the last half of the media was expelled forcefully to dislodge the aggregate but without spilling the medium. The aggregates were cultured for a further 24–72 h, after which 150 μL of media was removed and replaced with 150 μL fresh N2B27. From 96 to 120 h the media was replaced with fresh N2B27 supplemented with γ-secretase inhibitor DAPT at 50μM in order to inhibit Notch activity [43] or an equal amount of DMSO. At 120h the gastruloids in each condition were fixed.

## Human ESC culture and hNMP differentiation

The clinical grade hESC cell line MasterShef7 was used [89]. MasterShef7 hESCs were cultured on Geltrex (Gibco) coated wells in mTeSR Plus media. Routine hESCs passaging was performed with accutase in a 1:10 ratio once 70%–80% hESC confluency was reached, typically every 4–5 days. mTeSR Plus media was supplemented with ROCK inhibitor Y-27632 10μM for the first 24 h after passaging.

hESCs were differentiated to hNMP-LCs as described in [49,50]. hESCs were plated at 10,000 cells/cm$^2$ on Geltrex in N2B27 media supplemented with Y-27632 10μM with the desired concentration of recombinant FGF basic (R&D systems) and CHIR99021 (Axon). Typically, 20 ng/mL FGF and 2 or 3 μM CHIR99021 was used. After 24 h, the media was aspirated and fresh prewarmed N2B27/FGF/CHIR media without Y-27632 was added. hESCs were cultured for a further 48 h to acquire hNMP-LCs identity at 72 h.

## Immunohistochemistry

For E8.5 mouse embryos and cultured mouse embryos, the posterior portion from the posteriormost somite to the allantois were sub-dissected immediately prior to fixation. Embryo posterior portions and gastruloids were briefly washed in PBS and then fixed in 4% PFA in PBS + 0.1% Triton for 2 h at room temperature with gentle agitation. hNMP-LC monolayers were fixed by adding 8% PFA PBST to an equal volume of media retained in culture for a final 4% PFA concentration for 15 min. All tissues were washed in PBST and permeabilised in 0.5% Triton + PBS for 15 min, then placed in blocking solution at 4 °C overnight. All tissues were costained for SOX2 (Abcam, Ab92494), TBX6 (R&D Systems, AF4744), T-NL557 conjugate (R&D Systems, NL2085R), and LaminB1 (Abcam, Ab16048) conjugated to Alexa-Fluor647 using a conjugation kit per the instruction manual (Thermofisher, A20186). Sequential stains were performed by first staining SOX2 + TBX6 primaries, followed by donkey anti Rabbit Alexa Fluor 488 (Abcam, A21206) and donkey anti goat Alexa-Fluor594 (Invitrogen A21206) secondary for 24 h with gentle agitation. The secondary was blocked with 5% Rabbit + 10% Goat + 5% Donkey serum in PBST (PBST-RGD). The second primary stain with conjugated T-557 NL and LaminB1-647 were performed in PBST-RGD blocking solution. All primary and secondary stains were performed at room temperature with gentle agitation. 3D tissues were stained for 24 h and all primary antibodies were used at a dilution of 1:100.

Monolayer cell cultures were stained for at least 4 h and all primary antibodies were used at a dilution of 1:200. Three 15-min washing steps in PBST were performed at RT with gentle agitation between each step.

For hNMP high throughput single cell quantification, cells were routinely passaged and replated at half density onto Geltrex coated wells with media composition the same as the pre-passage culture media. Cells were left to adhere for 10–15 min and then immediately fixed as described above. The cells were stained with DAPI (Sigma, D9542) diluted 1:10,000 in blocking solution. Cells were stained as described above, except TBX6 secondary staining is performed with Donkey anti-goat Alexa-Fluor647 (Invitrogen, A21447). All secondary antibodies were used at a dilution of 1:1000.

## HCR RNA in situ hybridization: chick embryos

HCR RNA In Situ Hybridization (v3) [61] was carried out to analyze spatial patterns of gene expression in chick embryos. Embryos were dehydrated beforehand (after fixation) for storage at −20 °C until enough were obtained for HCR. This was performed using a series of 10 min graded methanol/PBST (PBS + 0.1% Tween) washes, starting with 2 × PBST, then 25% methanol, 50%, 75%, then 100% methanol followed by storage in 100% methanol. The reverse process (rehydration) was carried out before starting the HCR.

Rehydrated chicken embryos were placed in a 24-well plate. The PBST was removed and replaced with a 10 µg/mL solution of Proteinase K (Sigma, P4850) in PBST for 3 min at room temperature. This was replaced with 4% PFA for postfixing, for 20 min at room temperature. Embryos were then washed 2× with PBST, then washed once with 5× SSCT (SSC buffer, Thermo Scientific, J60839.K2, +0.1% Tween). All buffers used for HCR were made up according to the HCR RNA-FISH protocol (v3) for whole-mount chicken embryos. Pre-hybridization was carried out by applying Hybridization Buffer to the embryos for 30 min at 37 °C. Probes for Sox2, Mesogenin-1 (Molecular Instruments) and TBXT(Sigma-Aldrich) were diluted in Hybridization Buffer to 2 pmol and applied to the embryos, before incubating overnight (16–24 h) at 37 °C. The following day, probes were washed off using 4 × 20 min washes with Probe Wash Buffer at 37 °C, followed by 2 × 5 min washes with SSCT. Pre-amplification was carried out by washing with Amplification Buffer for 5 min on a shaker. Fluorescent hairpins (Molecular Instruments) were prepared by snap-cooling as per the HCR RNA-FISH protocol (v3) for whole-mount chicken embryos. Hairpins were then diluted to 30 pmol in Amplification Buffer. Pre-amplification solution was removed and replaced with hairpin solution, before incubating overnight at room temperature, in the dark on a shaker. The following day, hairpin solution was removed, and the embryos were washed 3 × 15 min with SSCT. Nuclei were stained with DAPI solution (1:1000 in SSCT) for 30 min at room temperature, then washed 3 × 10 min with SSCT.

## HCR in situ hybridization and antibody staining: *Drosophila* embryos

Prior to staining, fixed *Drosophila* embryos stored in methanol were rehydrated and permeabilised by a series of washes in 1 × PBS + 0.1% Tween-20 (PBT) rocking at room temperature as follows: 5 min each at 75%, 50%, and 25% methanol in PBT, then 3 × 10 min washes in PBT. HCR in situ hybridization was performed using v3.0 probes and hairpins produced by Molecular Instruments, following methods as described in Clark et al, 2022, with the exception that the 20 min postfix step following the HCR protocol was omitted. For antibody staining following HCR, embryos were incubated for 30 min rocking at room temperature in blocking solution (0.25% BSA [Sigma] in 5 × SSC + 0.1% Tween-20). Embryos were then incubated for 16 h in primary antibody diluted in blocking solution, at 4 °C with rocking. Embryos were washed 4 × 15 min in blocking solution at room temperature with rocking, then incubated for 16 h with fluorescently labeled secondary antibody diluted in blocking solution, at 4 °C with rocking. Embryos were washed 4 × 15 min in 5 × SSC + 0.1% Tween-20 (5 × SSCT), then incubated with 1 ng/µL DAPI (Thermo Fisher Scientific) in 5 × SSCT, for 30 min at room temperature with rocking, followed by 3 × 10 min washes in 5 × SSCT at room temperature with rocking. Primary antibodies were mouse anti-Lamin (Developmental Studies Hybridoma Bank ADL84.12) used at 1:50. Secondary antibodies were goat anti-mouse Alexa Fluor 700 (Invitrogen A-21036) used at 1:500. Prior to mounting, embryos were stored in 5 × SSCT at 4 °C.

## Imaging

Mouse embryos: Confocal microscopy was performed after tissue dehydration in a PBS/methanol series (5 min each) and two 10 min 100% ethanol steps. Dehydrated tissues were transferred to BABB (2:1 benzyl alcohol:benzyl benzoate) in ibidi μ-Slide wells for imaging. Monolayer tissues were mounted in 90% glycerol 10% PBST for imaging. Confocal imaging was performed on a Leica SP8 confocal microscope.

Chick embryos were placed on a glass coverslip within a border created with electrical tape. Excess SSCT was aspirated, then a drop of VectaShield (Vector Laboratories, H-1000) was placed on top of each embryo, before placing a second, smaller coverslip on top. The edges of the coverslip were sealed with nail varnish. Coverslips were stored at 4 °C prior to imaging.

*Drosophila* embryos were mounted in Prolong Gold Antifade Mountant (Thermo Fisher) between two high precision #1.5 coverslips (Thorlabs), enabling an embryo to be imaged from both sides. Microscopy was performed on a Leica Stellaris5 confocal microscope equipped with a white light laser (WLL). Images were acquired at 16-bit, with a 1,496 × 1,496 scan format and a 2 μs/pixel dwell time, using a Leica HC PL APO CS2 40×/1.30 Oil objective, a physical pixel size of 0.25 μm × 0.25 μm, and a z-stack step size of 0.29 μm. Each z-stack was specified to span from just above the top focal surface of the embryo, down past the center of the embryo and through to cells on the other side. This generated redundancy of slices around the center of the embryos between the image stacks taken from each side, enabling subsequent registering and stitching of the two image stacks, to reproduce the whole embryo. All imaging channels were acquired sequentially to minimize cross-talk. For each channel, the pinhole was set to 1 Airy Unit (AU), based on the peak emission wavelength for that fluorophore, with the exception of the DAPI channel where the pinhole was set to 2AU. The laser wavelengths and collection windows were: 405 laser and 415–480 nm window for DAPI; 498 nm WLL and 505–515 nm window for Alexa Fluor 488; 525 nm WLL and 535–545 nm window for Alexa Fluor 514; 559 nm WLL and 570–590 nm window for Alexa Fluor 546; 590 nm WLL and 610–630 nm window for Alexa Fluor 594; 653 nm WLL and 663–680 nm window for Alexa Fluor 647 and 685 nm WLL and 710–800 nm window for Alexa Fluor 700.

## Image analysis

3D Nuclei segmentation and curation: 3D single nuclei segmentation was performed using the Nuclear envelope segmentation system (NesSys) [3] using LaminB1 signal. Ten of the 19 segmented embryos were manually edited in PickCells to correct for segmentation errors. After manually editing the first 10 embryos, we used these as a ground-truth to establish whether manual editing actually made a difference to the final analysis. We established that it did not, and so we proceeded to analyze the remaining 9 embryos without manual correction.

High throughput single cell quantification: Image analysis on images from a high content confocal microscopy platform, including 2D segmentation and signal quantification, was carried out in the Signals Image Artist environment (Revvity Signals Software).

Neighbor identification was carried out in PickCells by Delaunay triangulation using nuclei centroids as input. For embryos, the epiblast was manually labeled in PickCells and only the epiblast nuclei were used as input for neighbor identification (see Fig 2A).

Nuclei to neighbor edges and individual nuclei features, were exported to R for all subsequent analysis including neighbor kernel smoothing, epiblast normalization procedure, and pseudospace identification.

Signal correction and normalization: All subsequent analysis was performed with custom scripts in Python. We first used linear unmixing to correct for signal bleed through between the TBXT and TBX6 channels. We also corrected for the loss of fluorescence intensity across the depth of the image due to light scattering effects. Next, raw values of mean signal intensity per nuclei were normalized by the formula (signal − background)/ (max signal − background). The max signal was chosen as the 99th percentile to exclude hot pixels and signal coming from staining artifacts such as cell debris and

antibody aggregates. The background signal was overrepresented in the image and was estimated by finding the highest peak in the signal histogram. Nuclei were classified as positive or negative by gating the populations manually after combining all the replicates per experiment.

Nuclear intensities smoothing: TF signal for each nuclei was iteratively smoothed by taking the average TF signal of the nuclei and its neighbors, normalizing the smoothed values using the same values used to normalize the raw values. This new average (AvTF + 1) was then used in another round, where the average AvTF signal of a nucleus was calculated (AvTF + 2). This was repeated 10 times (AvTF + 10) for each TF measured to spatially smooth the TF signal. After which, the log (AvTF + 10) signal was used as an input for PCA dimension reduction.

Pseudotime construction and NMP gating: The pseudo-space route from Neural to NMP to Mesoderm was identified using the Slingshot package [44] in PCA1 and PCA2 space. NMP region gates in pseudospace were ascertained by scanning gate values to best fit the bi-fated region in four SP embryos, which is then applied to all wild type embryo stages. Gates in pseudospace used to isolate NMP ROIs in ex vivo cultured embryos were identified by (i) the average pseudospace of node streak border nuclei and (ii) the average pseudospace values when mean population TBX6 rises above background along the pseudospace axis.

Defining a putative NMP region: We first allocate NMP-fated regions (black boxes on epiblast images) based on fate-mapping experiments that use microdissection and grafting of these regions to determine their fate [34,35]. We allocated the size and position of these regions using exactly the same approach as was originally used to identify regions for microdissection and grafting [34,35]: the width of the regions is set to match the width of the node, and the position of the boxes is determined in relation to the position of the node.

To define the U shaped putative NMP-like region, we identified pseudospace values corresponding to the known NMP-fated regions (i.e., for cells within the black boxes) and then searched for other regions with the same pseudospace identity: these other regions, together with the boxed regions, form the U shape corresponding to a region of NMP-like pseudospace identity.

To determine cells which should be excluded from our overall region of interest (excluded cells colored gray in Fig 3D and 3E) we first used our pseudotime analysis to find a spatial trajectory through from neural to mesoderm. We then took advantage of the fact that bipotent regions had previously been established based on grafting experiments [34,35], and set the posterior and pseudospace trajectory limits of the ROI based on this information. Some minor manual adjustment was used to ensure a coherent boundary (i.e., to avoid excluding sporadic nuclei that sat within the boundary).

The CV of TF signal intensity for a nucleus and its neighbors referred as the cell niche (CN) was defined as the standard deviation of TF signal intensities ($\sigma$) normalized by the mean of TF signal intensities ($\mu$) as below:

$$CN = \sigma/\mu \tag{1}$$

The NR was calculated as the nucleus' TF signal value ($I_{(nucleus)}$) divided by the average of its neighboring nuclei's TF signal intensities ($\mu_{(Neighbors)}$) defined as below:

$$NR = I_{(nucleus)}/\mu_{(Neighbours)}$$

TF values were simulated using the rnorm function in the compositions package in R. This function generates random numbers within a normal distribution, defined by a standard deviation and mean provided. In this case, the inputs for rnorm were calculated as in CV, using the standard deviation and mean of a nucleus's TF expression and the TF expression of its neighbors.

Earth movers distance [48] is a metric to measure the dissimilarity between two distributions. Here, it was calculated using R package https://cran.r-project.org/web/packages/emdist/index.html

## Epiblast machine learning classification

A Random Forest classifier was trained to distinguish epiblast from non-epiblast cells based on quantitative single-cell features. Each cell was characterized by local nuclear density, calculated as the number of neighboring nuclei within a fixed-radius 3D sphere; HCR signal intensities for the genes *TBXT*, *SOX2*, and *MSGN1*, computed as normalized fluorescence intensities per nucleus; and Euclidean distance from the dorsal-most 'surface' cell layer, defined as a surface interpolated from XYZ coordinates of the dorsal-most cell in 20 × 20 micrometer bins. Also, a mask of high density was created by segmenting creating a 2D image of average local nuclear density. Cells were clustered using k-means clusters to groups of ~5 cells. Average values per variable per cluster were used in the model. Features were not z-score normalized as this performed better here. A small subset of nuclei in XZ slices were manually labeled. Clusters containing manually labeled nuclei or surface cells within the high density mask were used to train the model and evaluate the model accuracy. Clusters of cells were labeled as epiblast or non-epiblast with ~99% accuracy. The Random Forest model was implemented using the scikit-learn library (version 1.6.1) with default parameters unless otherwise specified. Post processing to include missed nuclei included iterative rounds to convert non-epiblast nuclei if >50% of their neighbors are labeled as epiblast.

## Data representation and code

Figures were created with custom scripts in Python, R and MATLAB. Tools used to plot data include violin plots [90]. Color maps include the scientific colour map package scico (Crameri 2018, Scientific colour maps. Zenodo. https://doi.org/10.5281/ZENODO.1243909). Additional details of image analysis methods are provided in S1 Methods.

## Supporting information

**S1 Fig. 3D Manifold projection and alignment.** Example of the algorithm modules on a segmented six-somite pair embryo with the epiblast manually labeled. Points represent nuclei centroids and XYZ refer to coordinates in the raw image planes. In this process **(a)** first, the position of each nuclei in the anterior-posterior axis calculated relative to the nearest point on a principal curve in the Y and Z planes, which roughly correspond to the anterior/posterior (A/P) and dorsal/ventral axes, respectively. The dorsal/ventral axis is initially estimated as the distance to the nearest point on this A/P principal curve. The output of this **(b)** is carried to the next step (b) to identify the position along the left/right (L/R) and the dorsal/ventral (D/V) axis. In this step, sections of the epiblast parallel to the new A/P axis are isolated, four example sections are shown. In **(c)** Relative positions of nuclei to a principal curve in the X plane (roughly corresponds to the left/right axis) and the estimated D/V axis [distance to the A/P principal curve from (a) are calculated to identify the L/R and D/V axes. **(d)** The distances of each nuclei along the principal curve are normalized to the 'midline', which is the average L/R position of the 90th percentile of spatially smoothed TBXT values. Data for S1 Fig (A–D): Data file 1, https://doi.org/10.5281/zenodo.15802710. (DOCX)

**S2 Fig. Locating embryo landmarks.** Example of the manual marking of the posteriormost end of the notochord in a confocal stack of four somite pair embryo stained for LaminB1, SOX2, TBXT, and TBX6, with orthogonal views. (DOCX)

**S3 Fig. Mapping putative NMP regions.** Four somite pair embryo epiblast projections showing manual best-fits of pseudospace gates to bifated regions (in boxes). Showing left/right asymmetry in replicate three and non-specific bi-fated region labeling in replicate 5. Data for S3 Fig. Data file 3, https://doi.org/10.5281/zenodo.15802710. (DOCX)

**S4 Fig. 3D gradient neighbor calculation method.** An explanation of the method for calculating local gradient direction and steepness. Black arrows indicate positive values while red arrows indicate negative values. (DOCX)

**S5 Fig. Using ratios to describe relationship between two variables. (a)** Outline of scenarios to explain how log($x/y$) is suitable to describe the relationship between two variables. (i) In this scenario, $x/y$ does not equal $y/x$, and $y$ has logarithmic and asymmetric influence on the output of the calculation. (ii) Using log($x/y$) makes the calculation symmetric around 0, but the output is now non-linear. (iii) If the variables $x$ and $y$ can be assumed to be non-linear and a log transformation of $x$ and $y$ is appropriate, then the relationships between log($x$) and log($y$) with log($x/y$) is now linear due to the log law log($x/y$) = log($x$) − log($y$). **(b)** Summary of the scenarios and dynamic of the calculation output.
(DOCX)

**S6 Fig. Synthetic data. ( a)** Process to produce synthetic data. **(b)** Comparison of global population distribution for observed and synthetic SOX2, TBXT, and TBX6 signal intensity values. One four somite pair embryo dataset shown. **(c)** Density plots to compare observed and synthetic TF values of individual nuclei, showing close associations of SOX2 and TBXT, but less accurate TBX6 associations. All embryos shown $n = 19$. Data for S6 Fig (B–C): Data file 1, https://doi.org/10.5281/zenodo.15802710.
(DOCX)

**S7 Fig. Human ESC to NMP differentiation characterization and efficiency. (A)** High throughput single cell quantification method used to complement high resolution imaging (used in spatial neighbor analysis) by boosting sample sizes and N numbers. hNMP monolayers typically form dense structures which are difficult to segment and require time-consuming high-resolution images. To address this, cells are replated as single cells at a lower density and then quickly fixed to perform IF and stain for hNMP markers and DAPI. These are imaged on high imaging content platforms and present a simple challenge for single cell segmentation methods even at low resolutions. **(B)** Differentiation of human MShef7 to hNMPs with 20 ng bFGF and 2 μM or 3 μM CHIR (as described in Fig 6A) both produce high proportions of TBXT + SOX2+populations with no statistically significant difference between the groups. ($n = 5$). **(C)** CHIR titration of the hNMP differentiation protocol shows the influence of CHIR on SOX2, TBXT, and TBX6, where 3 μM CHIR produces a population with higher TBXT/TBX6 and lower SOX2 than 2 μM ($n = 9$). Points on violin superplot and barplot show mean per replicate. All error bars indicate confidence intervals of 0.95. Statistical tests performed by one way ANOVA, **$= p < 0.01$, ***$= p < 0.001$. Data for S7 Fig (B–C): Data file 2, https://doi.org/10.5281/zenodo.15802710.
(DOCX)

**S8 Fig. Methodology to identify NMP-like profiles in gastruloids using pseudospace. (A)** Process in identifying and excluding paraxial mesoderm-like clusters in gastruloids. (i) UMAP of TBXT/TBX6 patterning variables; namely normalized fluorescence units (NFI), mean neighbor NFI, local NFI heterogeneity (coefficient of variation). K-means clustering identifies clusters which when (ii) mapped onto the gastruloid localizes with paraxial mesoderm-like clusters. Graphic showing field of view for IF image and overlay of mesoderm-clusters and TF +ve cells. Mesoderm clusters are excluded in downstream analysis. **(B)** Similar in Fig 3, slingshot pseudotime tools are used in a PCA dimension reduction of smoothed normalized fluorescence units in gastruloids to identify pseudospace. NMP-like cells (NMPLC) are identified in gastruloids using pseudospace gates characterized E8 embryos. **(C)** Pseudospace values and NMPLC mapped onto the 10um section of a gastruloid using nuclei centroids. NMPLCs can be found not at the tip, but just 'anterior' of the tip in the elongation axis. Data for S8 Fig (B–C): Data file 2, https://doi.org/10.5281/zenodo.15802710.
(DOCX)

**S9 Fig. Steps for imaging and processing Drosophila embryos for PRINGLE.** 1. Embryo is mounted between coverslips and compressed to partially flatten to expedite image acquisition. 2. Embryos are imaged from both sides to generate two images from different views. 3. Each image is separately processed with single cell segmentation and quantification. 4. The two datasets are registered using the centroids of nuclei. 5. The nuclei with centroids in the half closest to the objective only are selected and carried forward for analysis.
(DOCX)

**S1 Methods. Document giving details of algorithms.**
(DOCX)

## Acknowledgments

We are grateful to Sophie Brumm and members of the Lowell, Wilson and Blin labs and to Linus Schumacher for helpful suggestions. We thank facility staff at the Institute for Regeneration and Repair of the University of Edinburgh, especially Justyna Cholewa-Waclaw at the High-Content Screening Facility, Matthieu Vermeren at the Imaging Facility, and Theresa O'Connor at the Tissue Culture Facility.

## Author contributions

**Conceptualization:** Matthew French, J. Kim Dale, Guillaume Blin, Valerie Wilson, Sally Lowell.

**Data curation:** Matthew French, Guillaume Blin, Valerie Wilson, Sally Lowell.

**Formal analysis:** Matthew French.

**Funding acquisition:** Benjamin Steventon, Erik Clark, Valerie Wilson, Sally Lowell.

**Investigation:** Matthew French, Rosa P. Migueles, Alexandra Neaverson, Aishani Chakraborty, Tom Pettini.

**Methodology:** Matthew French, Rosa P. Migueles, Alexandra Neaverson, Aishani Chakraborty, Tom Pettini, Guillaume Blin, Valerie Wilson.

**Project administration:** Sally Lowell.

**Resources:** Benjamin Steventon, Erik Clark, J. Kim Dale, Guillaume Blin, Valerie Wilson, Sally Lowell.

**Software:** Matthew French, Guillaume Blin.

**Supervision:** Benjamin Steventon, Erik Clark, J. Kim Dale, Guillaume Blin, Valerie Wilson, Sally Lowell.

**Validation:** Matthew French, Sally Lowell.

**Visualization:** Matthew French, Rosa P. Migueles, Aishani Chakraborty, Tom Pettini, Guillaume Blin.

**Writing – original draft:** Matthew French, Tom Pettini, Benjamin Steventon, Erik Clark, Guillaume Blin, Valerie Wilson, Sally Lowell.

**Writing – review & editing:** Matthew French, Tom Pettini, Benjamin Steventon, Erik Clark, Guillaume Blin, Valerie Wilson, Sally Lowell.

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
