## [Editor Report · Decision Letter 0]

Dear Sally,

Thank you for submitting your manuscript entitled "A toolkit for mapping cell identities in relation to neighbours reveals Notch-dependent heterogeneity within neuromesodermal progenitor populations" for consideration as a Methods and Resources by PLOS Biology.

Your manuscript has now been evaluated by the PLOS Biology editorial staff as well as by an academic editor with relevant expertise and I am writing to let you know that we would like to send your submission out for external peer review.

Once your full submission is complete, your paper will undergo a series of checks in preparation for peer review. After your manuscript has passed the checks it will be sent out for review. To provide the metadata for your submission, please Login to Editorial Manager (https://www.editorialmanager.com/pbiology) within two working days, i.e. by Sep 19 2024 11:59PM.

Kind regards,

Ines

--

Ines Alvarez-Garcia, PhD

Senior Editor

PLOS Biology

on behalf of

Suzanne De Bruijn, PhD,

Associate Editor

PLOS Biology

sbruijn@plos.org

---

## [Decision Letter · Decision Letter 1]

Dear Dr Lowell,

Thank you for your patience while your manuscript "A toolkit for mapping cell identities in relation to neighbours reveals Notch-dependent heterogeneity within neuromesodermal progenitor populations" was peer-reviewed at PLOS Biology. It has now been evaluated by the PLOS Biology editors, an Academic Editor with relevant expertise, and by several independent reviewers.

In light of the reviews, which you will find at the end of this email, we would like to invite you to revise the work to thoroughly address the reviewers' reports.

As you will see below, the reviewers think that this computational pipeline would be a valuable contribution to the field, however they also have concerns about the reproducibility and generalization. Both Reviewer #1 and Reviewer #4 mention that it would be useful to provide the pipeline in a more user-friendly format. Furthermore, both Reviewer #1 and Reviewer #3 raise concerns about the generalizability of the method to other systems beyond the fate transition analysed.

Given the extent of revision needed, we cannot make a decision about publication until we have seen the revised manuscript and your response to the reviewers' comments. Your revised manuscript is likely to be sent for further evaluation by all or a subset of the reviewers.

**IMPORTANT - SUBMITTING YOUR REVISION**

*Re-submission Checklist*

*Published Peer Review*

*PLOS Data Policy*

*Blot and Gel Data Policy*

Sincerely,

Suzanne

Suzanne De Bruijn, PhD,

Associate Editor

PLOS Biology

sbruijn@plos.org

REVIEWS:

*Reviewer #1: Identified himself as Jong Kyoung Kim

Summary: In this manuscript, the authors developed a computational pipeline to quantify and compare cell fate patterning from confocal immunofluorescence imaging data of developing embryos. Based on the tissue landmarks manually provided by the user, cells are positioned using relative coordinates, a key step in ensuring comparisons between embryos. They applied this pipeline to characterize the cell fate transition of neuromesodermal progenitors (NMPs) within the caudal lateral epiblasts of E8.5 embryos, and demonstrated that the patterning of NMPs and the observed heterogeneity of TBX6+ mesodermal derivatives are sensitive to Notch inhibition.

The present manuscript is well written and potentially important in providing a valuable computational pipeline for quantifying and comparing cell fate patterning within developing embryos. However, there are several major points that should be addressed to substantiate the authors' claims.

Major points:

1. The authors have nicely demonstrated the utility of their pipeline in the context of cell fate transition of NMPs within the caudal lateral epiblasts. However, as this is an ad hoc approach tailored to the caudal lateral epiblasts (requiring prior knowledge of the corresponding tissue landmarks), it is not clear how this approach is applicable to a more general context of developing embryos. For generalization, the authors should provide a more systematic approach to projecting cells onto relative coordinates based on tissue landmarks, and demonstrate that their approach can be applied to a more general context.

2. To increase usability and reproducibility, the code for the proposed pipeline should be available, carefully documented and wrapped into a R or python software package.

3. Figure 3C(ii), D, E: How and why the cells colored by gray were excluded?

4. Figure 3E: How the authors classify the cells into the three regions?

5. The CV values are usually inversely proportional to the mean values. Does this relationship affect the interpretation of Figures 5 and 6D?

Minor comments:

1. Figure 1C is not referred to in the manuscript.

2. "(Blin et al. 2019)" on Line 161-162 should be "[3]".

3. An unpaired parenthesis on Line 174.

4. Figure 2B: The tissue landmarks should be clearly marked in a schematic of the epiblst for readers not familiar with embryonic development.

*Reviewer #2: The manuscript entitled "A toolkit for mapping cell identities in relation to neighbours reveals Notch-dependent heterogeneity within neuromesodermal progenitor populations" presents a novel computational toolkit for characterizing cellular identity and patterning during embryonic development. The study focuses on the transition of neuromesodermal progenitors (NMPs) to mesodermal fate in E8.5 mouse embryos. The authors tackle the challenge of analyzing complex 3D tissue structures and cell fate transitions, using computational methods that quantify relationships between cells and their neighbors to understand the spatial patterning of cell identity.

The work makes several key contributions: First, it provides a faithful projection of 3D patterning into 2D that allows for comparisons of different embryos with each other in an unbiased way. Secondly, the authors show by analyzing the relationship between neighbourhood relations and cell identity that SOX2+ and TBXT+ cells tend to be surrounded by cells of similar identity, and TBX6-high cells are surrounded by TBX6-low cells. Finally, they developed a computation method called PRINGLE that is designed specifically to handle the complex curvature of 3D tissues.

Overall, the paper provides valuable insights into how neighboring cell interactions and signalling pathways contribute to the spatial patterning of cell fate decisions during embryonic development. The toolkit it presents will be useful for further studies on cell differentiation and tissue organization in both developmental and disease contexts. The results are scientifically sound and well-presented.

Minor points:

- In Figure 4c, the authors should state how they calculated the confidence intervals.

- In Figure 5b, the authors use Tukey's HSD to test whether the coefficient of variation of different transcription factors differs significantly. This test compares pair-wise means between groups but does not directly test for a difference in the coefficient of variation. The authors should use a test that directly addresses the coefficient of variation, such as Feltz and Miller 1996 or Krishnamoorthy and Lee 2014.

- In Figure 5f, the end points of each line have larger confidence intervals compared to the other data points. This suggests that the authors used a binning of pseudo-space to create this plot. If this is correct, they should state this in the caption including the bin size. The author should also indicate how they calculated the confidence intervals and in which form a potential binning of the data was considered for this.

- In the method section entitled "Nuclear intensities smoothing" the authors state that they run a PCA on the variable $\log{(AvTF+10)}$. They do not state whether they centered or scaled the data for this purpose. Does scaling or centering make a difference in the results?

*Reviewer #3: Pattern emergence and transformation is one of the key features encountered when studying developmental processes. Yet very few tools or analytic pipelines exist that allow for a quantitative approach to assess and compare features of the local microenvironment such as the expression of markers in the neighbouring cells and local coherence vs heterogeneity in patterning of cell states.

The work of Lowell and colleagues aims to bridge this gap as they describe a new pipeline that allows the comparison of patterns across multiple individual embryos/organoids or chosen ROIs in such structures. The authors used existing E 8.5 mouse embryo landmarks to compare emerging neighbourhoods' patterns between different embryos and subsequently used this pipeline to demonstrate that TBX6 positive cells in the NMP progenitor region tend to be surrounded by TBX6-negative cells and organised in a non-random distribution. By comparing patterns of the wt embryos with embryos that had been subjected to Notch inhibition, the authors found that discovered local heterogeneity of TBX6 depends on Notch signalling.

Importantly, the pipeline the authors proposed seems to be suitable to make in vivo/in vitro pattern comparisons, which in the opinion of this reviewer makes it an extremely valuable tool for any scientist interested in pattern formation and modelling of biological systems.

Before this elegant work is considered for publication, there are several points that need attention.

Major points

MM section - I am assuming that mice were naturally mated? Also, were all embryos collected from the same litter or several litters (I am assuming the latter is true but how many litters were analysed?) As the different shapes of the embryos were mentioned in the manuscript, have the authors checked how litter to litter variability influences the outcome of the PRINGLE method (and subsequent expression/gradient/neighbourhood etc analysis)? How similar were the results from the same stage embryos from different litters?

I am also unsure how simply using markers expression for the PRINGLE method to analyse gastruloids (data from fig 6) can work. As far as I can say, gastruloids often vary in shape, size and also expression pattern domains. How did the authors choose which expression patterns were the most representative/similar to the embryos? Alternatively, if the authors created an "average" projection using several gastruloids different in size, shape and/or expression pattern, how confident can we be that the final projection was representative/correct? Have the authors repeated the experiments several times and tested how similar are the projections from different experiments? Will these results be comparable between different labs (considering that efficiency of the whole process varies between labs and may influence expression domains?)

Although, I can clearly see the advantage of using the pipeline presented by the authors

I am not sure if gating the populations manually for positive/negative classification of the nuclei is the best strategy here. The authors made such a commendable effort to standardise most of the whole analysis and this specific step, in my opinion, introduces subjectivity to the whole process.

Minor points

There is an inconsistency of referencing through the whole text. The authors should decide on the referencing method and make it consistent in the whole manuscript.

It would be good to add E8.5 mouse embryos as nowhere in the title or abstract it is clear what systems the authors refer to.

Fig 2C - I am not sure how TBX6 staining on this panel corresponds to the staining presented in fig 1B (SOX2 and TBX staining look similar in both figures).

It would be good to explain what the difference between black and red arrows in Fig S4 3D is.

Fig 3 description is not finished (see line 246)

Fig 5D description, I am unsure what "solid lines" on the graph the authors refer to.

Fig 6 A description, What fig the authors had in mind by saying "(further characterisation fig SX)"?

I found the section titled "Local heterogeneity in TBX6 is sensitive to Notch inhibition" having not enough references to the fig 7, especially in the last 3 paragraphs. Adding appropriate references to the figure per each main finding, would help the reader to follow this section better.

In Fig 7F description, the authors wrote: "Note the statistically significant decrease in the CLE and NSB like regions for both TBXT and TBX6" but I cannot see TBXT data on that figure.

*Reviewer #4: Summary

The manuscript Matthew French et al. presents computational methods for quantitative analysis of cell patterning in complex 4D datasets. To this end, the authors focus on the axial progenitor region in mouse, containing neuromesodermal progenitors (NMP) expressing variable levels of different transcription factors across space and time. They combine previously developed and new computational pipelines to: 1) simplify visualization of 3D patterning in 2D, 2) define spatial domains from multiple averaged samples, 3) use cell-neighbor relationships to define marker expression changes and mesoscale patterning, and 4) apply this to various experimental conditions. One of the relevant findings, facilitated by their cell-neighbor analysis, is that TBX6 expressing cells in the NMP domain have a non-random distribution. They suggest a potential lateral inhibition mechanism for mesoderm commitment in this region.

I particularly value the development of pipelines to analyze and visualize 3D tissue patterning across averaged samples and experimental conditions. This work is of significant value for the field and will facilitate the study of tissue patterning from developmental biology to disease modeling. However, I have major questions and concerns, related with the explanation of the methods, interpretation of some results and the comparison of this method with existing ones, that will need to be addressed prior to publication. My specific comments are included below.

Major comments

1. A major concern is that the methods presented here are not accessible to non-experts and cannot be reproduced with the information currently provided. These are my suggestions:

o Provide all the code through an Open Access platform. Reference the algorithms presented in Supplementary Material to the corresponding code.

o Reduce the use of jargon and explain why a particular method was used to non-experts. For example, in line 361: "We then computed the earth move distance between the observed…" the concept was not explained before and there is no further information in supplementary information.

2. An important goal of this manuscript is to avoid arbitrary selection of regions based on marker thresholding. However, there are multiple examples of manual selection of regions or cell selection:

o Line 166: "The epiblast is manually isolated using PickCells software". Please show how you define the region of interest in different embryos.

o Line 174: "…node (manually labelled)…". While the authors provided one example in Figure S2, it would be best to show how they select the same region across embryos to avoid introducing bias.

o Line 705: " Ten of the 19 segmented embryos were manually edited in PickCells to correct for segmentation errors." Please explain why only ten were manually edited.

o Line 722: " Nuclei were classified as positive or negative by gating the populations manually after combining all the replicates". Please explain why this needs to be done manually or provide alternatives to classify the cells automatically.

3. Previous computational methods were developed for analysis of 3D and 4D datasets across samples (References18-31). One example is MorphoGraphX 2.0 (Ref 27) which includes a software interface, while others have custom pipelines to analyze patterning across averaged samples (Ref 31). To demonstrate the significance of the new method in this manuscript, it would be important to compare their analysis with at least one of these existing methods. If none of the previous methods allows for this type of analysis, please discuss why.

4. The use of pseudotime tools to represent pseudospace is an interesting approach (Figure 3). However, the conclusion that NMP region extends beyond the known fate maps needs clarification. The authors should explain how they allocate fate map regions (black boxes) from previous studies to their own epiblast projections. Further, they should explain how they define a U-shaped region from pseudospace and those allocated bi-fated regions - are there thresholds? They did not explain any of this in the text or Methods.

5. The number of samples analyzed for some developmental stages should be increased to extract representative conclusions. Specifically, there were only 2 samples analyzed at somite pair stage 10-12.

6. I have major concerns related to the Notch inhibition experiments:

o The authors claim that Notch manipulation does not affect shape and size of the cultured epiblast (Line 486). However, Figure 7B clearly shows size differences between control and treated conditions. If this is true, an explanation about how to apply their projections and analysis methods to compare both conditions is needed.

o The authors claim that there is a bilateral symmetry of the embryo in this region and they combine both sides into single projections (Line 489 and Figure 7D-F). This is not consistent with their previous demonstration of left-right asymmetry (Figure S3, Line 791). They should repeat the analysis separating left and right sides.

o The relationship between Notch and TBXT6 has been previously demonstrated in different contexts (https://doi.org/10.1002/gene.20124;
https://doi:10.1101/gad.1248604 ; https://doi.org/10.1073/pnas.0508238103 ). The authors should cite and discuss prior knowledge and how this relates to their findings.

o Based on this and the fact that their observations are still quite preliminary and do not allow to conclude whether Notch signaling is the source of local heterogeneity in the NMP, I think the Title should not include Notch.

Minor comments

Introduction:

1. The authors should try to improve the citations of previous related literature. For example, in Lines 59-63 they cite some methods for 3D data analysis across samples but there are others (https://doi.org/10.7554/eLife.78300).

2. Line 67: "…known regulators of cell lineage, and has a well-defined fate map". A reference is missing here.

Results and Figures:

3. Line 118: "It is not easy to determine by eye whether these sporadic TBX6+ cells are restricted to a particular location or expressed in a particular pattern". There are previous methods that allow to not do this by eye. Please comment on why those previous quantitative computational approaches were not sufficient for this.

4. Line 162 (Blin et al.) should be a Reference number instead.

5. Lines 209-211: on the comment"…this marker-based approach presents challenges because both TBXT and SOX2 are expressed in graded distributions, meaning that gating for positive cells can be somewhat arbitrary." There are methods allowing non arbitrary selection of thresholds, such as 'Mixture analysis' (Reference 9). Please discuss why this is not enough and the authors still use manual gating for cell selection.

6. Figure 3 Legend. Add reference in (B). Clarify how you obtained (E) from (D).

7. Line 265. Please add a reference with specific examples.

8. Figure 4C: please improve visualization of the curves with respect to the shading area. Potentially change the background color for the plots to help visualize this.

9. Line 332: please provide a more specific example Reference.

10. Line 344: here the analysis considers cell-neighbors heterogeneity. Please explain how this can be averaged between different samples where cell resolution is lost.

11. Figure 5F: please change the colors between Simulated and Observed so that they can be distinguished better.

12. Line 364: the authors claim that TBXT local heterogeneity corresponds to a random distribution. However, the data in figure 5G (ii) shows a significant difference in distribution for TBXT in the transition zone between NM and Mesoderm. Please explain.

13. Line 411: the authors use hES derived monolayers, and they should explain why they chose a different species instead of mouse. Could this be another source of variability?

14. Figure 6C: SOX2 also presents higher variability in the density than the other factors. The authors should comment on that result.

15. Line 499: substitute "T" by "TBXT".

16. Figure 7F: adding a Supplementary Figure showing some examples of the original data used for this figure showing TBX6 distribution changes between controls and treated samples would be helpful to correlate this 100-cell grid patterns to the original data.

17. Figure S6 (b). Please use colors that can help distinguish between Observed and Synthetic curves.

Discussion:

18. Line 546: the statement "comparison between samples at the single cell level" is not accurate given the averaging and comparisons are not done at that resolution.

19. Please discuss how this method overcomes previous methods challenges (see Major comments).

20. Please comment on potential limitations of your study and suggest ways to improve it in the future.

Methods:

21. Line 661: is 36C incubation needed? (37?).

22. Line 665: what is DAPT? Please provide references for reagents.

23. Line 683-686: please provide the concentration used for the different antibodies.

24. Line 708: please specify the confocal microscopy platform used.

---

## [Decision Letter · Decision Letter 2]

Dear Sally,

Thank you for your patience while we considered your revised manuscript "A toolkit for mapping cell identities in relation to neighbours reveals conserved patterning of neuromesodermal progenitor populations" for publication as a Methods and Resources article at PLOS Biology. This revised version of your manuscript has been evaluated by the PLOS Biology editors, the Academic Editor and by three of the original reviewers.

The reviewers are largely satisfied by the revision and we are therefore likely to accept this manuscript for publication. However, before we can accept your study, we would like to invite a last, short revision so that you can address the remaining point from reviewer 3, as well as the following editorial requests.

**IMPORTANT: please address the following editorial requests, in addition to reviewer 3's last concern

1. FINANCIAL DISCLOSURES: Please indicate, in your financial disclosures statement in our online system, whether the sponsors or funders of your study played any role in the study design, data collection and analysis, decision to publish, or preparation of the manuscript.

2. ETHICS STATEMENT: Please update the methods section of your paper to include the full name of the IACUC/ethics committee that reviewed and approved the animal care and use protocol/permit/project license. Please also include an approval number.

3. DATA AVAILABILITY: You may be aware of the PLOS Data Policy, which requires that all data be made available without restriction: http://journals.plos.org/plosbiology/s/data-availability. For more information, please also see this editorial: http://dx.doi.org/10.1371/journal.pbio.1001797

a. Supplementary files (e.g., excel). Please ensure that all data files are uploaded as 'Supporting Information' and are invariably referred to (in the manuscript, figure legends, and the Description field when uploading your files) using the following format verbatim: S1 Data, S2 Data, etc. Multiple panels of a single or even several figures can be included as multiple sheets in one excel file that is saved using exactly the following convention: S1_Data.xlsx (using an underscore).

b. Deposition in a publicly available repository. Please also provide the accession code or a reviewer link so that we may view your data before publication.

>>Regardless of the method selected, please ensure that you provide the individual numerical values that underlie the summary data displayed in the following figure panels as they are essential for readers to assess your analysis and to reproduce it:

Fig 2C; 3A-E; 4B-C; 5B-D,F-G; Fig 6C-D; Fig 7D-F; FIg 8 D-I; Fig 9 C-H

Fig S1A-D; Fig S3; Fig S6B-C; Fig S7B-C; FIg S8B-C;

>>Please also ensure that figure legends in your manuscript include information on where the underlying data can be found, and ensure your supplemental data file/s has a legend.

>>Please ensure that your Data Statement in the submission system accurately describes where your data can be found.

4. CODE: Thank you for providing the code generated for this study on GitHub. While this is fine to include, please note that we cannot accept sole deposition of code in GitHub, as this could be changed after publication. Instead, we need you to archive this version of your publicly available GitHub code to Zenodo. Once you do this, it will generate a DOI number, which you will need to provide in the Data Accessibility Statement (you are welcome to also provide the GitHub access information). See the process for doing this here: https://docs.github.com/en/repositories/archiving-a-github-repository/referencing-and-citing-content

We expect to receive your revised manuscript within two weeks.

*Published Peer Review History*

*Press*

Sincerely,

Luke

Lucas Smith, Ph.D.

Senior Editor

lsmith@plos.org

PLOS Biology

Reviewer remarks:

Reviewer #1, Jong Kyoung Kim (note, reviewer 1 has signed this review): The authors have satisfactorily addressed my comments.

Reviewer #3: The authors answered satisfactory all my main points. Providing additional analysis of chick and Drosophila embryos significantly improved the manuscript and demonstrated that the the pipeline described in the manuscript can be employed to many diverse systems.

I have only one minor comment. Figure 8 legend needs to be revised. By the look of it, panel F and G description was swapped/mixed up

Reviewer #4: I have now read the Revised manuscript, and I feel it is now acceptable for publication, as the authors have clearly addressed all my concerns and made important changes to the manuscript.

---

## [Editor Report · Decision Letter 3]

Dear Sally,

Thank you for the submission of your revised Methods and Resources article, "A toolkit for mapping cell identities in relation to neighbours reveals conserved patterning of neuromesodermal progenitor populations" for publication in PLOS Biology, and thank you for addressing the last editorial and reviewer requests in this revision. On behalf of my colleagues and the Academic Editor, Bon-Kyoung Koo, I am pleased to say that we can in principle accept your manuscript for publication, provided you address any remaining formatting and reporting issues. These will be detailed in an email you should receive within 2-3 business days from our colleagues in the journal operations team; no action is required from you until then. Please note that we will not be able to formally accept your manuscript and schedule it for publication until you have completed any requested changes.

PRESS

Sincerely, 

Luke

Lucas Smith, Ph.D.

Senior Editor

PLOS Biology

lsmith@plos.org